# Decomposing Prediction Mechanisms for In-Context Recall

## Abstract

We introduce a new family of toy problems to explore challenges with long context learning and associative recall in transformer models. Our setup involves interleaved segments of observations from randomly drawn linear deterministic dynamical systems. Each system is associated with a discrete symbolic label that must be learned in-context since these associations randomly shuffle between training instances.

Via out-of-distribution experiments we find that learned next-token prediction for this toy problem involves at least two separate mechanisms. One "label-based" mechanism uses the discrete symbolic labels to do the associative recall required to predict the start of a resumption of a previously seen system's observations. The second "observation-based" mechanism largely ignores the discrete symbolic labels and performs a prediction based on the state observations previously seen in context. These two mechanisms have different learning dynamics: the second mechanism develops much earlier than the first.

The behavior of our toy model suggested concrete experiments that we performed with OLMo training checkpoints on an ICL translation task. We see a similar phenomenon: the model learns to continue a translation task in-context earlier than it decisively learns to in-context identify the meaning of a symbolic label telling it to translate.

## 1 Introduction

The release of GPT-3 (Brown et al., 2020a) demonstrated the power of Large Language Models' (LLMs) ability to do in-context learning (ICL). Since then, there has been significant progress in understanding ICL for language models themselves (Olsson et al., 2022; Akyürek et al., 2024; Xie et al., 2021; Lin & Lee, 2024; Wei et al., 2023b; Yin & Steinhardt, 2025; Wies et al., 2023; Pan et al., 2023; Min et al., 2022). Nonetheless, practitioners still find that LLMs have idiosyncratic biases in their text generation, exhibit brittleness with under-specified prompts, and perform poorly with long-context when the task requires tracking multiple streams of information (Shao, 2024; Zheng et al., 2024; Hsieh et al., 2024).

There has also been work that focuses on understanding ICL for simpler toy problems (Garg et al., 2022; Rajaraman et al., 2024; Edelman et al., 2024; Singh et al., 2025; Raventós et al., 2024; Du et al., 2023; Nichani et al., 2024). Problems such as linear regression (Garg et al., 2022; Raventós et al., 2024; Huang & Ge, 2025) allow us to study the learned ICL behavior of deep neural networks in settings where optimal strategies are known, allowing complex prediction mechanisms to be disentangled. In this paper, we build on previous work to create a new toy problem involving interleaved vector-valued time-series, that provides a simple setting that exhibits the token-level sensitivity of ICL and allows for experiments that test the prediction mechanisms of models trained for next-token prediction.

### 1.1 Basic setup

We start with underlying time-series that come from the evolution of random deterministic linear systems and thus these time-series play the role of noise-free least-squares problems in Garg et al. (2022) — each consecutive time-series observation is defined by its underlying deterministic linear

system (specified by an unknown matrix, just as in linear regression). This continuous-state problem has a naturally continuous error metric: mean-squared-error, and therefore avoids interpretive challenges that may come with discrete error metrics Schaeffer et al. (2024). As in Garg et al. (2022), ICL for predicting observations from a dynamical system implicitly involves identifying the underlying system from observations of its evolution.

Segments of random length from different time-series are interleaved with discrete "symbolic punctuation labels (SPL)," tokens that unambiguously label different segments as belonging to different time-series. These SPLs and the fact that they can occur repeatedly within the context window introduces a dimension of recall similar to multi-query-associative-recall (MQAR) (Arora et al., 2023) or multi-queries needle-in-a-haystack (MQ-NIAH) (Hsieh et al., 2024). However, successful recall is not simply a matter of copying a particular surface-level value from the context. Instead, the corresponding task (predicting the next observation in this particular sequence) must be done.

The discrete symbolic nature (in the sense of (Wei et al., 2023a)) of these punctuation labels means that their meanings must be learned in context to be able to complete associative recall — *they cannot be memorized as the associations change for every new instance of the problem*. Similarly, the details of each distinct time-series itself must also be learned in context. We restrict attention to noiseless time-series defined by orthogonal matrices — consequently, once we have seen enough information in the observations for a specific time-series, in principle, perfect prediction accuracy is possible.

To be specific about our terminology in this work, we will call each training or test example a *trace*. This trace is made of interleaved *segments* of observations from distinct dynamical systems. Each segment is labeled with an SPL corresponding to the dynamical *system* that generates the segment. Details of the setup are in Section 2 and Fig. 2 shows an example of a training trace.

## 1.2 CONTRIBUTIONS

We examine two abilities required to complete our task (1) the ability to *identify* a segment from the corresponding SPL, (here corresponding to predicting the token at the first index after the query), and (2) the ability to *continue* predictions for a segment after having one observation from the system (here corresponding to predicting tokens at indices 2, 3 ... after the query). These correspond to abilities that transformer models must perform for natural language tasks as well. Consider an in-context translation task where the model must learn the target language for the translation in context, based on a symbolic label. Then, the model must in-context learn to (1) *identify* the target language and (2) *continue* the translation. Our contributions focus on the distinct mechanisms by which the compound prediction task is performed by the transformer model.

**Distinct emergence for two abilities:** We find clear evidence of the emergence of associative recall during training in our toy problem (see black curve in Fig. 1b). Furthermore, we find that the *continue* ability for the model emerges before the *identify* ability (see blue and red curves).

**Falsify label-based recall hypothesis:** A natural hypotheses for the recall mechanism is:

> **H1: Label-based recall.** The model uses in-context learning of the association of symbolic labels to systems, and then performs inference based on recalling the queried system and continuing its evolution.

Surprisingly, we find that H1 is not uniformly true for the toy model. We perform a simple misdirection experiment (see second row of Fig. 1a) where the test sequence uses the wrong SPL for a particular segment (for example, where we insert the SPL for system-A, even though the observations that follow the SPL come from some other system, say system-B). Under H1, we would expect that such misdirection would lead to poor performance for several indices following the misdirected SPL. We see this for the first index following the query, i.e. the 1-after query index. Notice the sharp rise in the black error curve in Fig. 1c, at exactly the training checkpoint where the ability to identify the segment based on the SPL emerges in Fig. 1b.

However, we see in Fig. 1c that the ability to perform 2-after query and 3-after query index predictions seems to be unaffected by this misdirection, with blue and red curves identical to the normal setup in Fig. 1b. The curves seem to behave as though the model were ignoring the symbolic labels entirely. This surprising observation leads to our next hypothesis.

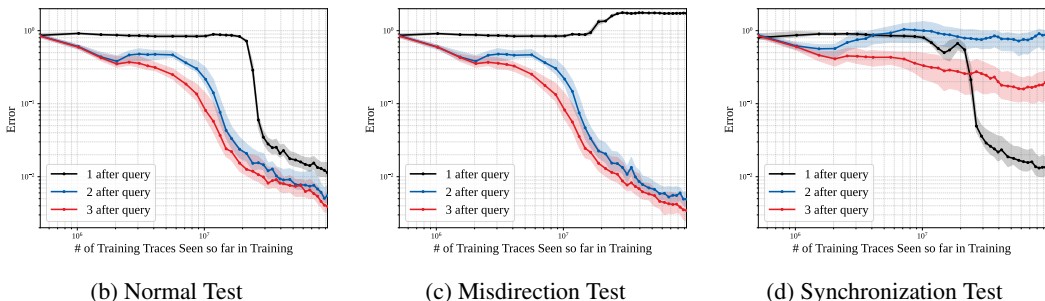

(a) The testing setup and the key misdirection and synchronization experiments. A normal test trace uses the correct SPL (denoted here by an open parenthesis) as the query symbol following the 'haystack'. The misdirection experiment inserts an SPL that does not correspond to the segment following it. The synchronization experiment uses the correct query but renders the first index following the query uninformative for predicting the subsequent index in the segment.

(b) Normal Test          (c) Misdirection Test          (d) Synchronization Test

Figure 1: Training dynamics for normal recall, and two out-of-distribution (OOD) experiments. The curves show the prediction performance for indices 1 (black), 2 (blue), and 3 (red) steps after the Query symbol. *We find that transformers can have distinct learning dynamics for an ICL task at different positions over the task!* Pure recall (the black curve in Fig. 1b) emerges suddenly, while later positions' recall (blue and red curves) develops earlier. Through two OOD experiments, misdirection (Fig. 1c) and synchronization (Fig. 1d), we further isolate the distinction of the mechanisms behind initiating an instance of the ICL-task (black curves) vs. continuing it (other curves).

**Falsify observation-based recall hypothesis:** We then consider the complementary hypothesis,

> **H2: Observation-based recall.** The model ignores the symbolic labels. Once the model sees a new observation, it compares it with previously seen observations to figure out which previously seen system it could have come from.

We know from the misdirection experiment that H2 is also false as a complete explanation, since the 1-after query prediction does depend on the SPL. But is H2 true for the subsequent indices? We further explore this through a backwards synchronization experiment (described in Sec. 3.2.2) which makes the 1-after-query index uninformative for what the 2-after and 3-after query indices should be (results in Fig. 1d). Here we see that the 1-after-query performance (black curve) does not degrade, but the performance for 2-after (blue) does degrade significantly. This further supports the idea that the 1-after-query prediction follows H1, while the mechanism for 2-after and beyond follows something like H2. However, it cannot be a perfect version of H2 because the 3-after (red) query curve in Fig. 1d also shows severe degradation even though there is no observation-level ambiguity — seeing $U_1 \mathbf{x}_{10}$ information-theoretically reveals that system $U_1$ must be active rather that $U_2$. See Appendix D for a formal presentation of these hypotheses.

**Observation-based recall degrades more with context length than label-based recall:**   As the context length increases, we see that the model's recall performance degrades for those indices that we believe are using a largely observation-based mechanism while holding steady for the index using label-based recall (see Fig. 3).

**Distinct circuits for distinct tasks:**   A further edge-pruning based investigation shows that the circuits for predicting the 1-after query and 2-after query indices are completely distinct.

**A conjecture:**   Our observations lead us to the following conjecture:

> **C3: Transformers develop distinct mechanisms for predicting different token positions in a single task.** Distinct mechanisms are used to initiate a new episode of an ICL task (i.e., predict the first index after query) versus continuing that task (i.e., predict the second index after query).

**Confirmed conjecture in a natural language task:**   We modify a classic LLM emergence experiment (Wei et al., 2023b;a) to create a simple in-context translation task as described earlier. Using OLMo checkpoints (OLMo et al., 2024), we confirm that this conjecture C3 holds for this task — i.e. even before the emergence of successful initiation of an ICL-specified task, models can successfully continue performing that task.

## 2   SETUP

Consider predicting the continuous-state of an *unknown* linear dynamical system. We focus on the orthogonally evolved system family (Sander et al., 2024), where the system is defined by $U \in \mathbb{R}^{5 \times 5}$, a random orthogonal matrix. Each $U$ is generated by the algorithm presented in (Mezzadri, 2006), which ensures a uniform sampling over all $\mathbb{R}^{5 \times 5}$ orthogonal matrices. The initial state is $\mathbf{x}_0 \sim \mathcal{N}\left(0, \frac{1}{5}I\right)$, with state updates:

$$\mathbf{x}_{i+1} = U\mathbf{x}_i = U^{i+1}\mathbf{x}_0. \tag{1}$$

The system state is in-principle perfectly predictable, but only after six positions in the sequence are observed by solving for

$$U = \begin{bmatrix} \mathbf{x}_1 & \mathbf{x}_2 & \mathbf{x}_3 & \mathbf{x}_4 & \mathbf{x}_5 \end{bmatrix} \begin{bmatrix} \mathbf{x}_0 & \mathbf{x}_1 & \mathbf{x}_2 & \mathbf{x}_3 & \mathbf{x}_4 \end{bmatrix}^{-1}. \tag{2}$$

Due to this fact, an optimal algorithm for learning the underlying system from state observations is to use the pseudoinverse in place of the inverse as presented in Appendix C.1.

**Cutting and interleaving training sequences**   To form a training trace, we interleave segments of observation sequences from a library of 40,000 orthogonal systems into a length-251 context window. The exact interleaving procedure is described in Appendix C.2. The maximum number of systems in a trace is sampled from a $\mathrm{Zipf}(1.5, 25)$ distribution.

Note that within a single training example, segments of a particular system always start with the same open token and always end with its corresponding close token. These random assignments are redrawn at the beginning of the interleaving process for each training example; therefore, *the same system can have different symbolic open and close tokens when it appears in different training examples*. This ensures that there is no semantic prior from pretraining for these open and close tokens and forces their local meaning to be learned in-context (Wei et al., 2023b). See Fig. 2 for a diagram of an interleaved training example.

**Needle-in-a-haystack test**   For testing, we generate 100 held-out systems and 1000 different held-out initial states[1] by the same method described in Section C.2 to form our *testing library*. Since the testing systems do not appear in the training set, if the model is able to achieve low error on testing systems, then it must be able to in-context learn new orthogonal systems.

We create a series of structured "needle-in-a-haystack" test traces by interleaving traces in the testing library. Each "needle-in-a-haystack" trace with $N$ systems is generated by inserting a segment of

---

[1]We generate 1000 initial states for each system to narrow down the quartiles in the squared-error curves.

$$\text{<start>} \begin{bmatrix} \mathbf{x}_0^{(30)} & U_{30}\mathbf{x}_0^{(30)} & \cdots & U_{30}^{24}\mathbf{x}_0^{(30)} \end{bmatrix} \left\{ \mathbf{x}_0^{(2)} \quad U_2\mathbf{x}_0^{(2)} \quad \cdots \quad U_2^4\mathbf{x}_0^{(2)} \right\} \left\{ U_2^5\mathbf{x}_0^{(2)} \quad \cdots \quad U_2^{26}\mathbf{x}_0^{(2)} \right\} \left( \mathbf{x}_0^{(771)} \cdots U_{771}^{45}\mathbf{x}_0^{(771)} \right) \begin{bmatrix} U_{30}^{25}\mathbf{x}_0^{(30)} & \cdots & U_{30}^{166}\mathbf{x}_0^{(30)} \end{bmatrix}$$

Figure 2: Example of a 251-element-long interleaved training example. The different colors correspond to the different segments that are being interleaved. In this example, the purple sequence is generated from system 30 in the training library (denoted in the superscript of the initial state and the subscript of the $U$ matrix), while the blue sequence is generated from system 2. Notice that each system in the training trace has its own pair of SPLs represented by parentheses, curly braces, and brackets in the figure. Lastly, notice that when a sequence from a system that already appears in the trace gets added, the current sequence continues from where the previous sequence ended, as seen by the last purple segment starting from index 25 since the first purple segment ended at index 24, and the last blue segment starting at index 5 since the first blue segment ended at index 4.

10 state observations starting from index 0 from the testing library into the test trace (the haystack). Each of these segments are individually punctuated with a unique open and close symbol pair. We then append a query open symbol to the test trace (corresponding to the needle), followed by 10 state observations that continue the system corresponding to the query open symbol (the test segment).

A more detailed version of this procedure is in the Appendix E.1.2. See the top row of Fig. 1a for a diagram of a test trace for $N = 2$ systems in the haystack and system $U_1$ as the needle.

## 3 RESULTS

### 3.1 ONLY OBSERVATION-BASED RECALL DEGRADES WITH MORE SYSTEMS IN THE HAYSTACK

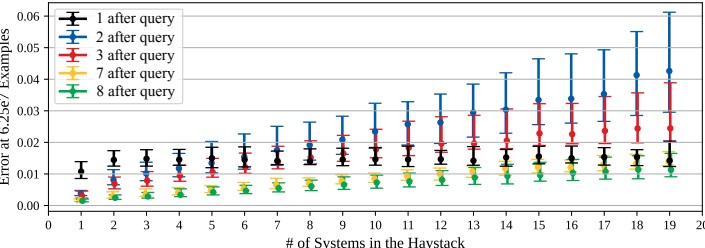

Figure 3: The 25th, 50th, and 75th quartiles of the squared-error after $6.25 \times 10^7$ training examples as the number of systems in the haystack $N$ increases. Notice that predicting 1-after the query is largely unaffected by the number of systems in the haystack, as the black markers stay steady around 0.015. Although the information encoded in the open symbolic label keeps the difficulty of the task constant as the number of systems increases, nevertheless, the performance for 2,3,7, and 8-after the query symbol predictions steadily degrades as the number of systems increases.

In Fig. 3, we plot the quartiles of the squared-error of the model's predictions as a function of the number of systems in the haystack. Observe that the 1-after query performance remains steady with more systems, while the predictions for the later indices get progressively worse. Predictions on one index after the query achieve an MSE of $\approx 0.015$ which is significantly below the MSE of 0.87 that is achieved by the optimal pseudoinverse predictor from Appendix C.1 when it predicts the first index of a sequence (see Appendix Fig. 16a). This shows that the model is able to successfully perform in-context recall and that this kind of label-based recall does not degrade with context size.

Information theoretically, making a prediction for 2-after the query is just as easy as making a prediction for 1-after the query, if not easier, since the query label provides the information required to make an optimal prediction. In light of this, Fig. 3 suggests that the model does not learn to use symbolic labels well enough to maintain steady performance on the indices two or more after the query as the number of systems in the haystack increases, supporting the multi-mechanism conjecture and suggesting that observation-based recall is more vulnerable to degradation with context size.

Intriguingly, Hsieh et al. (2024) found that the Yi model (a 34B parameter LLM claiming to have a context length of 200,000 tokens) also struggled to perform the needle-in-a-haystack task when there were an increasing number of "distractor" needles in the haystack. These distractor needles are the analog of the other systems in the haystack that are not being queried in our toy. At first glance, this finding from Hsieh et al. (2024) might seem to suggest that the needle-in-a-haystack task is more difficult when there are more possible needles for the model to choose from. But the steady performance of the 1-after query predictions as the size of the haystack increases for our toy model falsifies this claim at least in our toy setting. Furthermore, our toy setting uncovers a new hypothesis: the degradation in the Yi model's performance could be due to insufficient development of the symbolic recall ability and the over-development of observation-based recall during its pretraining.

## 3.2 Out-of-Distribution Inference-Time Experiments

We conducted three out-of-distribution experiments at inference-time to test H1 and H2: misdirecting the model towards the incorrect sequence in the haystack (Section 3.2.1), synchronizing sequences in the haystack from different systems so that they would all have the same state after the query open symbol (Section 3.2.2), and misdirecting the model towards an unseen sequence not present in the haystack (Appendix F.3). To further explore the mechanism for restarting ICL on a new system, we conduct a fourth out-of-distribution experiment: misdirecting the model towards a seen sequence in the haystack (Appendix F.4).

### 3.2.1 Experiment 1: Misdirection towards the incorrect sequence in the haystack

For the misdirection experiment, we test the model on "needle-in-a-haystack" test traces except we swap the query open symbol with another open symbol that corresponds to a segment in the haystack that is not the "needle" (see Fig. 1a). If H1 were true, the model would be using label-based recall and we would expect it to make predictions on the test segment for the wrong system. If H2 were true, the swapping of the label would not affect the prediction performance of the model, since the model would be using the seen states to make its predictions rather than the symbolic labels.

Misdirecting the model towards the incorrect sequence in the haystack falsifies pure label-based recall and provides strong evidence that an observation-based recall mechanism is present. The results of this experiment are shown in Fig. 1c with $N = 2$. This figure plots the median squared-error of the model's predictions against the number of training examples seen so far. The first thing we notice is that the solid black curve now sharply rises late in training, as opposed to its sharp fall as the associative recall ability emerges for a normal "needle-in-a-haystack" test trace as shown in Fig. 1b. This is replicated in the Appendix for other values of $N$ and more detailed figures are provided in Appendix Fig. 19. This supports H1: the model seems to be using label-based recall to predict the first index of the test segment.

In contrast, the red and blue curves in Fig. 1c for 2 and 3, after the query open symbol look almost identical to the corresponding solid curves in Fig. 1b. For example, see the blue curve in Fig. 1c and 1b at $2 \times 10^7$ training examples. Their median squared-error both sit at around $2 \times 10^{-2}$. The same correspondence can be seen for the red, yellow, and green curves in Appendix Fig. 19 as well. This shows that the model's predictions for 2, 3, 7, and 8 after the query open symbol are not very affected by the open symbol, and are likely using the state observations after the query open symbol to decide which system to use for making predictions. This strengthens H2, that the model uses an observation-based recall mechanism for predicting tokens after the first index into the test segment.

### 3.2.2 Experiment 2: Synchronizing "rotations" in the haystack

The misdirection towards the incorrect sequence experiment showed that the model can make accurate predictions for indices 2 and further into the test segment without using the query open symbol. However, in that setting, information about which system is in the test segment is also present in the first observation after the query symbol, and this can determine the system to be applied for predicting the subsequent indices. Say the first symbol after the open query symbol is $\mathbf{x}_{10}$. Then, if a predictor has last observed $\mathbf{x}_9^{(1)}$ from system 1 and $\mathbf{x}_9^{(2)}$ from system 2, it can check whether $\mathbf{x}_{10} = U_1 \mathbf{x}_9^{(1)}$ or $\mathbf{x}_{10} = U_2 \mathbf{x}_9^{(2)}$, and thereby identify if the system is 1 or 2.

Therefore, a question remains. Can the model can use the query open symbol to predict the later indices in a situation where the 1-after-query observation does not provide the necessary information? To answer this question, we conduct a synchronizing rotations experiment. We create a test trace where all of the sequences in the haystack from different systems have identical states at the $10^{\text{th}}$ index, which corresponds to the first index after the query open symbol. To do this, we first generate a single vector $x_{10} \sim \mathcal{N}\left(0, \frac{1}{5}I\right)$ for all systems and generate the haystack by "rewinding" our systems back to their initial state $x_0$ by $x_{i-1} = U^T x_i$ as is shown in Fig. 1a in the synchronization experiment. To resolve the resulting ambiguity, a model must use the symbolic label to make an accurate prediction on the second index into the test segment.

Synchronizing the sequences in the haystack so the observation at the first index in the test segment is not informative of the continuing system shows that the model has not learned to use the symbolic label to make accurate predictions for two or more indices into the test segment. In Fig. 1d, for $N = 2$, we have the median squared-error of the predictions on the synchronizing test traces vs. the number of training examples seen so far, we see that the solid black curve (that shows the performance of the 1-after query index) still sharply decreases late in training[2], as it does in Fig. 1b, showing that the model is able to use the query open symbol to predict $\mathbf{x}_{10}$. On the other hand, the solid blue curve for the 2-after query is almost horizontal throughout all of training, as is the red curve for 3-after query. This means that the model is unable to make accurate predictions on the subsequent indices in the test segment after $\mathbf{x}_{10}$, although the query open symbol provides all of the necessary information to do so.

In the more detailed figures in Appendix Fig. 20, we see similar phenomenon for 7 and 8 indices after the query, as well as the same behavior for $N = 5$ systems. This strengthens the case for H2 as the right hypothesis for predicting the indices after $\mathbf{x}_{10}$. It is worth noting that seeing the 1-after symbol $\mathbf{x}_{10}$ and 2-after symbol $U\mathbf{x}_{10}$ can allow the model to completely identify $U$, but it does not seem to be doing this given the performance on the 3-after query index. This can be seen in the red solid curve in Fig. 1d being still much worse than the counterpart curve in Fig. 1b.

### 3.2.3 SUMMARY OF OUT-OF-DISTRIBUTION EXPERIMENT RESULTS

After conducting the two out-of-distribution experiments, we find that neither H1 nor H2 can fully explain the model's mechanism for predicting interleaved time-series. The misdirection to the incorrect sequence experiment (Section 3.2.1) supports H1 for predicting the first index into the test segment, but it also shows that the model can make accurately continue its predictions in the test segment without using the query open symbolic label. Therefore, H1 is insufficient. The synchronizing rotations in the haystack experiment (Section 3.2.2) further showed that the model is unable to use the query open symbolic label well enough to accurately continue its predictions in the test segment, providing strong evidence that the model performs a suboptimal version of H2 for this subtask. In further OOD experiments, this evidence is strengthened by examining the models behavior during misdirection towards a seen sequence in Appendix F.4. The experiment to misdirect to an unseen sequence (Appendix F.3) shows that even for the later indices, the query open symbolic label can signal the model to restart its predictions for a new segment, invalidating H2 as the sole mechanism for continuing predictions. These results support conjecture C3 — the transformer model uses multiple mechanisms for the single task of interleaved time-series prediction.

### 3.3 MECHANISTIC ANALYSIS ON PREDICTION MECHANISMS

Now that it is clear the model uses multiple mechanisms for the single task of interleaved time-series prediction, we study if these different mechanisms have different computation graphs in the weights of the learned transformer model. To do this we use Edge Pruning, a transformer circuit-discovery method that optimizes over continuous masks over a disentangled transformer to find a sparse representation of a task (Bhaskar et al., 2024). We run a modified version of Edge Pruning to distinguish the circuits being used by our model for the 1-after query and 2-after query tasks. As our model is solely trained with squared-error, we remove the KL objective in the original Edge Pruning method's loss function in (Bhaskar et al., 2024) and optimize on a scaled up squared-error added to the original edge loss. The loss function that we optimize to find a sparse computation graph is $\mathcal{L}' = k \cdot \mathcal{L}_{\text{squared-error}} + \mathcal{L}_{\text{edge},s}$. Our dataset for the edge pruning method consisted of

---

[2]Due to the synchronization, the first observation in the test segment is a valid continuation for all systems in the haystack and therefore is always predictable.

one "needle-in-a-haystack" trace configuration generated in the same way as in Appendix E.1.2 for 5 systems in the haystack.[3]

| Circuit | # Edges | 1-After squared-error | 2-After squared-error |
|---|---|---|---|
| Orthogonal Sys Full Model | 32936 | 0.002 | 0.004 |
| Orthogonal Sys 1-after Circuit | 200 | 0.01 | 0.64 |
| Orthogonal Sys 2-after Circuit | 40 | 0.32 | 0.02 |

Table 1: Edge pruning finds sparse transformer circuits with high evaluation accuracy in our GPT2-style model. We prune late checkpoints of a model using interleaved traces and data consisting of 5 systems in the haystack. We report the number of edges in the final circuit and the mean squared-error of the circuits' predictions for both the 1-after query and 2-after query tasks. We run an edge thresholding binary search to reach the target edge sparsity, set all pruned edges to have weights of 0, and then run inference. We visualize the 1-after query circuit in Appendix Fig. 25.

We report the size of the circuits and the squared-error of the predictions outputted from both the final checkpoint of the trained model and the pruned circuits in Table 1.[4] Since the pruning method optimizes continuous gates for each node in the computation graph, these continuous gates must be quantized to 0 or 1 to get a pruned circuit (Bhaskar et al., 2024). Our results in Table 1 show the accuracy of the pruned model *after* this quantizing has been done, which differs from how (Bhaskar et al., 2024) reports their results. Post-quantization performance ensures that the edges that we believe to be irrelevant truly have no contribution to the output. Importantly, we find high accuracy and *0% edge overlap* between the 1-after query and 2-after query circuits, indicating that *our model mostly leverages mechanistically different learned mechanisms for consecutive tokens*. See Appendix G for further details on the pruned circuits.

## 4  DO PRETRAINED LLMS ALSO DISPLAY MULTI-MECHANISM TENDENCIES?

To see whether our conjecture (multiple mechanisms for a single multi-token task) holds for natural language problems solvable by prompting LLMs, we leverage OLMo-2 7B checkpoints (OLMo et al., 2024) and a basic English to Spanish translation task that is inspired[5] by the IPA translation task used in previous works benchmarking and studying emergent behaviors (Wei et al., 2022; bench authors, 2023). As our symbolic labels in the dynamical system tasks correspond to different systems during training, we also change our analogous natural language setup to have in-context labels with no semantic meaning, similar to Wei et al. (2023a). In Fig. 4b, we see a similar phase transition in the first token prediction task, a parallel of the 1-after-query dynamics of the associative recall setup (compare to the black curve in Fig. 1b). Meanwhile, the second-token performance is both better and more gradual in its improvement across training. This qualitatively matches the toy problem behavior in Fig. 1b.

To verify that Fig. 4b is not the result of a difficulty gap between the first token task compared to the second, we replace the purely symbolic task labels "X:" and "Y:" in the few-shot examples with the original semantically informative "Spanish:" and "English:" labels. This replacement converts the problem from pure ICL for task recognition (*in-context associative recall*) to leveraging a previously learnt label (*in-weights associative recall*). The resulting performance is seen in Fig. 4a. Notice the marked improvement in the first-token performance that erases the entire gap to the second-token performance. This establishes that the model steadily learns how to start a translation over pretraining, it just can't in-context-learn how to use the symbolic label until the phase transition around 50k training steps. Given the true underspecification of user tasks in the wild (Zhao et al., 2024), and the

---

[3]Further experiments should test more "needle-in-a-haystack" configurations, and would have a larger testing library of traces to create full train and test splits for training and evaluating the pruning method.

[4]The trained model that was used for these results is from an earlier training run than the "orthogonal medium" model that is throughout the rest of this paper. This earlier training run was trained on $5 \times 5$ orthogonal matrices that were sampled from an (unintentionally) non-uniform distribution over all $5 \times 5$ orthogonal matrices (Mezzadri, 2006).

[5]We use Spanish instead of the International Phonetic Language (IPA) as IPA has tokens that are not compatible with the OLMo 2 tokenizer. Examples of the English to Spanish task are shown in Appendix H.

common dependence of semantic labels in benchmarks (bench authors, 2023), this marked phase transition could indicate a source of commonly seen benchmark to user performance differences.

We also explore a variant to the misdirection OOD task (Appendix F.3). Here, we consider the task where the symbolic label that is used for directing the associative recall is replaced with a different symbolic label that is so far unseen in context. To adapt this task to a natural language setup, we keep the translation task with two few-shot examples exactly the same as Fig. 4b, but alter the final testing prompt to not use the 'Y:' label and to instead use a previously unseen 'Z:' label (Examples in Appendix H). We see the results in Fig. 4c. Immediately, it is obvious that the second-token-prediction task is functionally unaffected by the "misdirection" by the previously unseen token 'Z:'. This gives further evidence that the second token prediction does not rely on the in-context label in order to complete the task once it has started. This matches what we saw in our toy model as well, lending further support for our multi-mechanism conjecture to hold in pretrained LLMs. Meanwhile, for the first token task (black curve in Fig. 4c), in a sense, there is objectively no right answer for what to do when presented with 'Z:'. Nonetheless, we measure accuracy against a correct ground-truth English translation. The model accuracy oscillates wildly as it trains, which is an unexpected behavior for single-epoch training and deserves more investigation in future work.

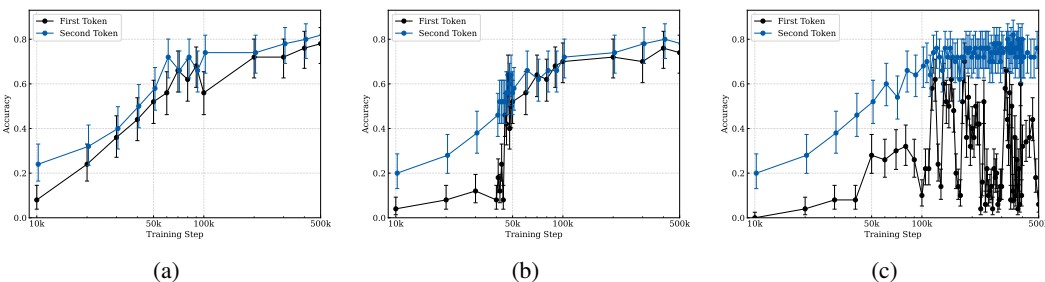

(a)          (b)          (c)

Figure 4: Comparative example of in-weights associative recall (a), in-context associative recall (b), and an in-context associative recall task that is misdirected to an unseen label (c), all in a 2-shot prompting setting. Each point is a separate OLMo-2 7B model at different training steps. We report 95% credible intervals using Jeffreys prior ($Beta(0.5, 0.5)$) based on 100 samples per evaluation point. Similar to the toy model, we see that the ability to apply an in-context recognized task is emergent, after which the performance of the first output token increases to the level of other output tokens. We also see that misdirection to an unseen label does not effect the second token accuracy when compared to the original task.

## 5 DISCUSSION

At this point, the community understands that ICL is rich and nuanced (Lin & Lee, 2024; Wang et al., 2024; Min et al., 2022; Park et al., 2025; Lampinen et al., 2024). We contribute a new dimension of nuance by empirically pointing out that *a single ICL-driven task can be performed, on different tokens, using multiple mechanisms that emerge separately*. When tasks have tight local coherency, there can often be approximate local underspecification — there are multiple ways of knowing what the model is supposed to be doing here. The intrinsic Bayesian orientation of autoregressive next-token prediction (Xie et al., 2021) means that this local underspecification can get picked up on, while the average-loss-over-tokens driving the training gradients means that an approximately correct mechanism that works most of the time will be rewarded and improved even if it can't solve the task completely. The very success of this approximately correct mechanism during training will further reduce the overall gradient pressure (Pezeshki et al., 2021) for alternative and potentially better mechanisms, potentially forcing them to develop more slowly (Shah et al., 2020). However, structurally, there are certain aspects of tasks that are likely to require the use of the better mechanisms — and it seems that starting an episode of a task might be one of them. These better mechanisms therefore can emerge later in training — but their emergence does not mean that the better information they are acting on will automatically be incorporated by the mechanisms already favored for other parts of the task. This contrasts with situations where the earlier emerging mechanism foregrounds information that can be used by the later mechanism (Singh et al., 2025).

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

## A    NEW FIGURES FOR RESPONSES TO REVIEWERS

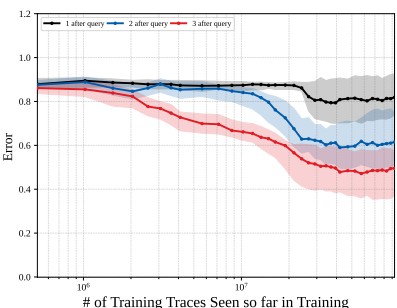

(a) Two entries per haystack segment. On linear scale for clarity.

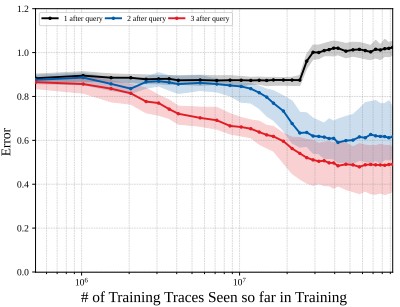

(b) Misdirection to wrong sequence. On linear scale for clarity.

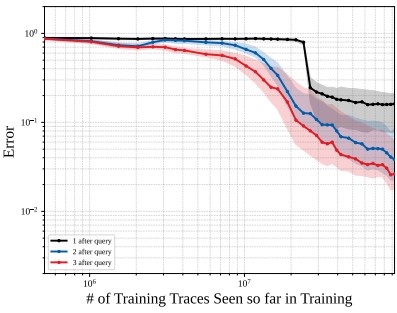

(c) Five entries per haystack segment.

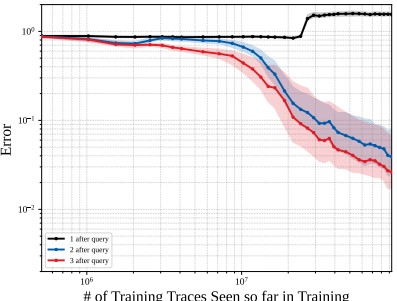

(d) Misdirection to wrong sequence.

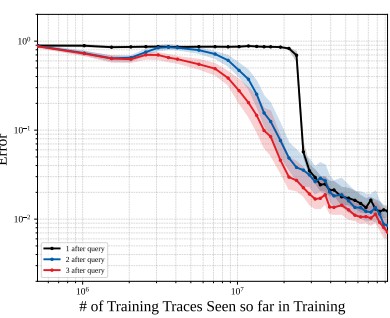

(e) 10 entries per haystack segment.

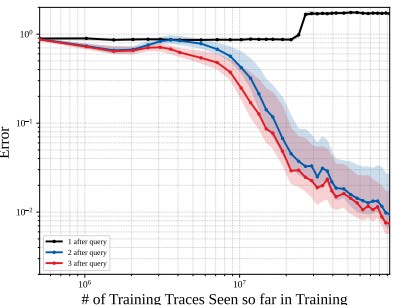

(f) Misdirection to wrong sequence.

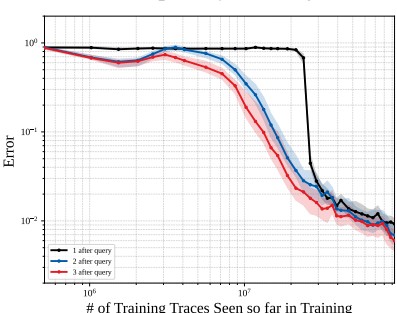

(g) 15 entries per haystack segment.

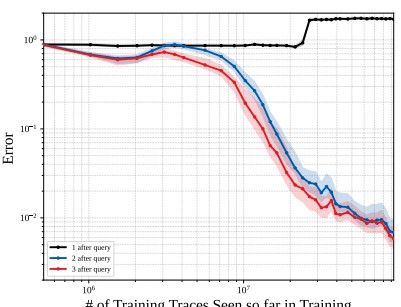

(h) Misdirection to wrong sequence.

Figure 5: Squared-error vs. number of training points seen so far during training for various lengths of haystack segments. Five systems in the haystack.

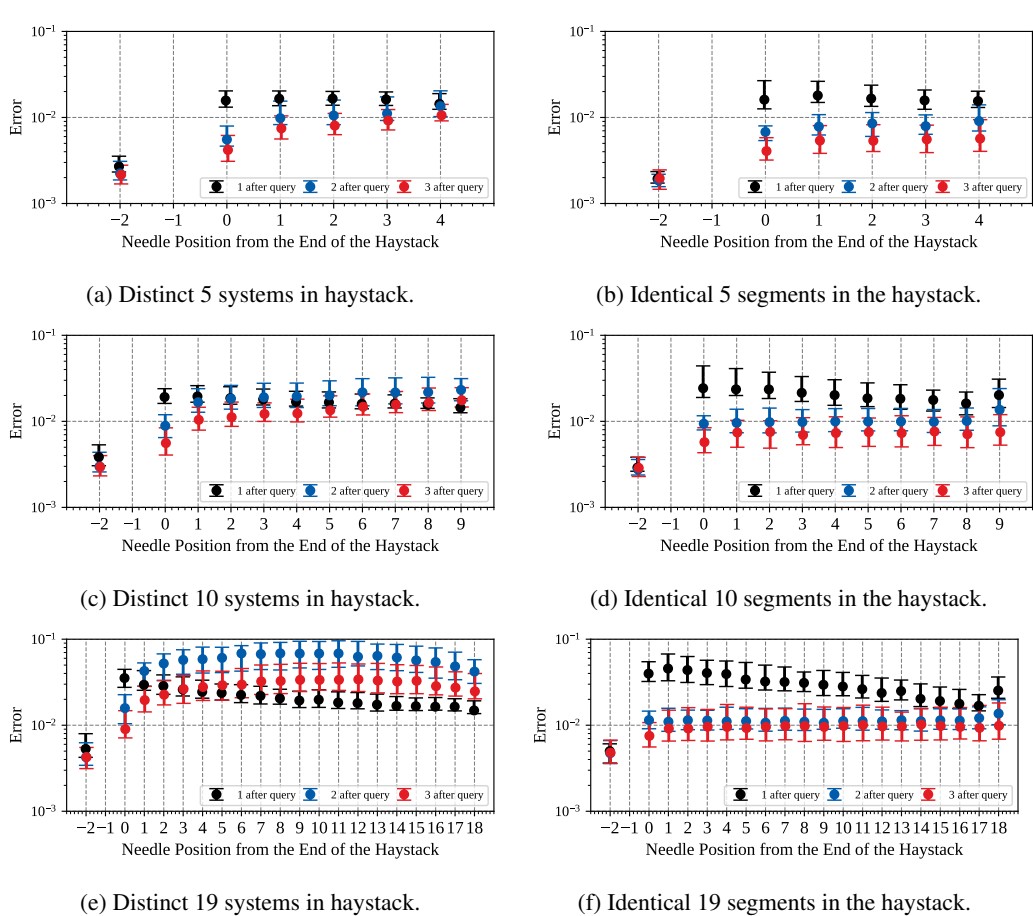

(a) Distinct 5 systems in haystack.

(b) Identical 5 segments in the haystack.

(c) Distinct 10 systems in haystack.

(d) Identical 10 segments in the haystack.

(e) Distinct 19 systems in haystack.

(f) Identical 19 segments in the haystack.

Figure 6: Squared-error for 1,2, and 3 steps after the query vs. the position of the segment in the haystack that is being recalled (needle position). All haystack segments have 10 entries. For these plots, the x-axis value of $-2$ corresponds to a test segment that is the continuation of the segment before with no interruption by open and close symbols. The x-axis value of 0 is the closest needle position to the test segment while the largest x-axis value is the first segment seen in the context window. This is at the checkpoint where $6.25 \times 10^7$ training examples have been seen so far during training.

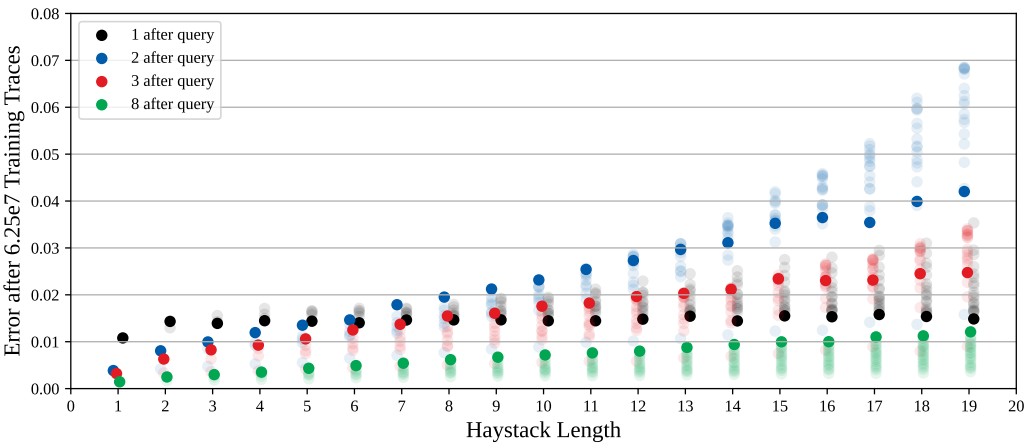

(a) Normal needle-in-a-haystack test with distinct systems.

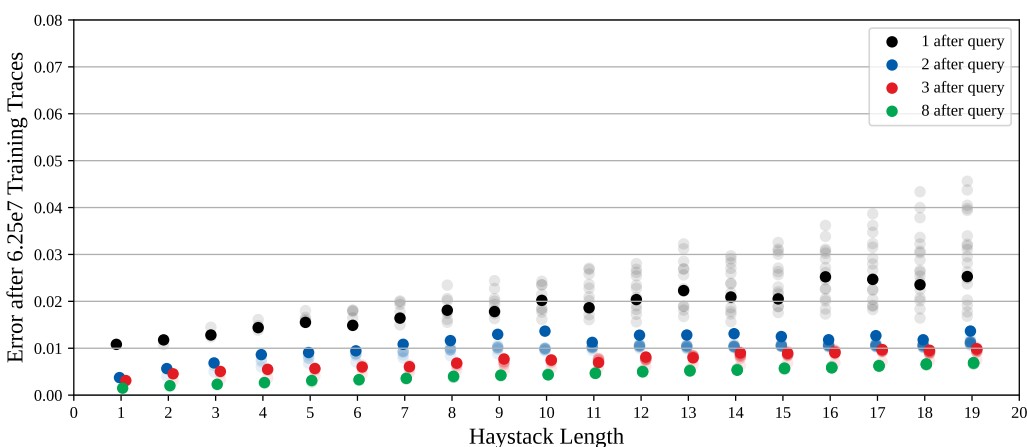

(b) Identical haystack test with identical traces.

Figure 7: Squared-error vs. number of systems in the haystack for all positions of the needle within the haystack. The needle at the first segment seen in context is the darkest dot while the other needle positions are lighter.

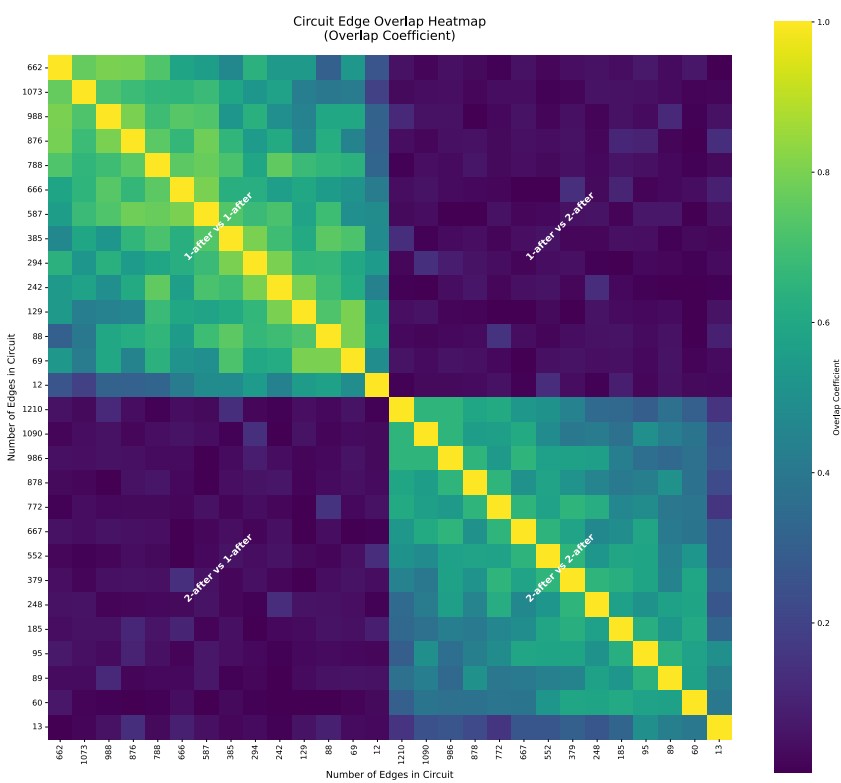

Figure 8: A heatmap showing the similarity between the pruned 1-after and 2-after circuits over 14 trials of edge pruning. The top 14 rows and the left 14 columns correspond to the 1-after circuits and the remaining rows and columns correspond to the 2-after circuits. The tick labels show the number of edges found in each pruned circuit: ranging from around 1000 to closer to 10 depending on the setting of the edge-pruning hyperparameter that controls sparsity.

## B  EXTENDED RELATED WORK

**Emergence**  Benchmark performance of large language models (LLMs) has been observed to improve abruptly at certain scales in a seemingly unpredictable manner (Wei et al., 2022; Ganguli et al., 2022), leading to the conclusion that LLMs exhibit *emergent* abilities, where different abilities may emerge at different scales or points during training. A natural question was posed by (Schaeffer et al., 2024): is emergence a mirage due to the discrete nature of the evaluation metrics used? Would alternative continuous metrics (e.g. logits instead of generated responses) show continuous improvement and thus lead to more predictable performance? Or are these phase transitions real? At this point, the reality of phase-transitions in abilities during training is well established, with arguably the strongest evidence of this coming from the "grokking" literature (Power et al., 2022; Nanda et al., 2023; Gromov, 2023; Zhong et al., 2024; Mallinar et al., 2024; Humayun et al., 2024; Mohamadi et al., 2023; Soudry et al., 2018; Liu et al., 2022; Prieto et al., 2025; Pezeshki et al., 2022; Davies et al., 2023) where there is a clear emergence of substantially improved generalization performance after what looks like a long period without improvement during training. Similar behavior has been observed in stylized regression-style examples that are very different from LLMs and ICL (Lyu et al., 2023; Kumar et al., 2024; Nam et al., 2024). The toy problem in our paper is both regression-style and very much related to ICL and LLMs.

What exactly drives emergence is also an ongoing topic of investigation. While earlier work talked about model sizes and total compute, the story now is more nuanced. Powerful evidence connects the emergence of abilities to the pretraining losses attained (Du et al., 2025), other information-oriented metrics (Chen et al., 2024b), and the idea that more complex or specialized abilities can only emerge after a model acquires prerequisite abilities during training (Chen et al., 2024a; Lubana et al., 2024).

**In-context learning**  Our understanding of in-context learning (ICL) has greatly benefited from experiments in simple domains, with many works studying ICL through linear dynamics (Garg et al., 2022; Huang & Ge, 2025; Wu et al., 2024). To add a proxy for the order-dependence of language, works have used sequences from Markov chains to study pretraining, revealing emergent behaviors related to the formation of induction heads (Edelman et al., 2024; Sander et al., 2024; Du et al., 2023; Li et al., 2023).

Recent works have analyzed the role of labels in ICL, giving additional perspective into information flow and the inherent limitations of ICL (Min et al., 2022; Wang et al., 2023; Huang & Ge, 2025). This is closely linked to the literature on in-weights learning (IWL) (Chan et al., 2025; Anand et al., 2025), which reveals structural contrasts in the learning mechanisms models can leverage.

Mechanistic explorations of ICL have advanced through both top-down classification of higher-order attention head behaviors (Elhage et al., 2021; Olsson et al., 2022; Yin & Steinhardt, 2025) and bottom-up analyses tracing the dynamics of these features (Singh et al., 2024). From analyzing the learning dynamics of ICL in various setups (Mainali & Teixeira, 2025; Zhang et al., 2025), works have found that ICL enables a mixture of emergent behaviors (Reddy, 2023; Chan et al., 2022; Nguyen & Reddy, 2024; Lu et al., 2023) and competes with other learning mechanisms (Park et al., 2025). Contemporaneous work has also explored the transient dynamics of ICL (Singh et al., 2025), finding that a form of IWL both cooperates and competes with ICL in attention-only models.

With learning dynamics displaying various surprising phenomena in different setups, works have also tackled discerning whether there are distinct modes of ICL (e.g. task learning (TL) vs. task recognition (TR)[6]) through various lenses like ICL risk (premature convergence), competition dynamics, and differing data distributions (Pan et al., 2023; Wang et al., 2024; Lin & Lee, 2024; Wies et al., 2023). Other work has argued for broader definitions than the "dual mode hypothesis", pushing for ICL to be understood as a spectrum of behaviors or mixture of algorithms (Lampinen et al., 2024; Park et al., 2025). In a sense, our findings here add another nuance to this story since we provide evidence that different modes of ICL can be active at the same time for the same task, just on different tokens, and that these modes can emerge separately during training.

---

[6]From (Wang et al., 2024): "TR refers to the ability of an LLM to recognize the target task from demonstrations and only utilize its own knowledge obtained from pretraining to solve the task, while TL refers to the ability of an LLM to solve the target task solely based on demonstrations." The NLP example we choose in Section 4 of this paper falls into the "Task Recognition" subtype.

**Associative recall** With the resurgence of state-space models, (Arora et al., 2023) highlighted associative recall (Hopfield, 1982; Zhang & Zhou, 2017) as the key capability where attention-based models hold a distinct advantage. This was formalized by introducing a discrete toy problem they called Multi-Query Associative Recall (MQAR). However, in basic MQAR, the solution involves the exact copying of the right token from context rather than recalling something conceptual that has to be applied locally. Meanwhile, in continuous Gauss-Markov models, next-observation prediction can still be performed effectively using state-space models, as least-squares algorithms can be adapted into a streaming form via standard Woodbury-matrix tricks. By using MQAR-style discrete symbolic labels to segment time-series drawn from Gauss-Markov models, we effectively create a naturally continuous toy problem where the information to be recalled is not an exact copying.

**Are deep neural networks Bayesian predictors?** The Bayesian perspective on machine-learning generally is longstanding and consequently has been used to better understand the ICL capabilities of deep neural networks. (Xie et al., 2021) frames in-context learning as implicit Bayesian inference. Using a simple, synthetic task setup, this work showed theoretically that if a model makes its predictions according to its pretraining distribution, it will asymptotically approach the performance of the Bayes optimal predictor as the number of in-context examples for this task increase to infinity. (Panwar et al., 2024) builds off of these ideas, and empirically investigates how close a transformer model's predictions are to the Bayes optimal predictor. They find that high-capacity transformers successfully perform Bayes-optimal prediction for Gaussian mixtures and other more complex mixtures. Interestingly, they hypothesize that low-capacity transformers attempt to approximate Bayesian inference through gradient descent. The facts that gradient descent reaches the global optimum for convex problems and (Von Oswald et al., 2023) shows that multiple self-attention layers in a model can simulate multiple steps of gradient descent lead the authors in (Panwar et al., 2024) to believe that higher-capacity transformers perform Bayesian inference because they can take more steps of gradient descent.

Many works have shown that gradient descent can approximate a Bayes-optimal predictor. For example, (Mandt et al., 2017) uses stochastic gradient descent with a constant learning rate to sample from an approximate posterior, by connecting stochastic optimization with the stochastic gradients used in Markov chain Monte Carlo. Additionally, (Mingard et al., 2021) uses Gaussian processes to estimate a posterior and empirically computes the posterior probability of a deep neural network trained with stochastic gradient descent. Through comparing the two posterior distributions, they conclude that the Bayes posterior is a first-order correlate with the neural network's, while second-order differences can come from hyperparameter choices when training the model.

Relating this work to the emergence that we observe for transformer models on our in-context recall task, singular learning theory predicts that Bayesian learners have phase transitions in their generalization error (Watanabe, 2009). Grounding their empirical investigation in this theory, (Hoogland et al., 2025; Chen et al., 2023) study the loss landscape of transformer and autoencoder models, and are able to predict the occurrence of a phase transition by computing the local learning coefficient (Lau et al., 2024).

## C EXTENDED SETUP

Consider predicting the continuous-state of an *unknown* linear dynamical system. We focus[7] on the orthogonally evolved system family (Sander et al., 2024), where the system is defined by $U \in \mathbb{R}^{5 \times 5}$, a random orthogonal matrix. Each $U$ is generated by the algorithm presented in (Mezzadri, 2006), which ensures a uniform sampling over all $\mathbb{R}^{5 \times 5}$ orthogonal matrices. The initial state is $\mathbf{x}_0 \sim \mathcal{N}\left(0, \frac{1}{5}I\right)$, with state updates:

$$\mathbf{x}_{i+1} = U\mathbf{x}_i = U^{i+1}\mathbf{x}_0. \tag{4}$$

---

[7] In the Appendix, we provide results for the identity system family which has dynamics that are even simpler than the orthogonal system family. For the identity systems, the initial state is $\mathbf{x}_0 \sim \mathcal{N}\left(0, \frac{1}{5}I\right) \in \mathbb{R}^5$. Now, the state updates as

$$\mathbf{x}_{i+1} = \mathbf{x}_i. \tag{3}$$

This trivial process of copying a constant is perfectly predictable after one realization is observed.

The system state is in-principle perfectly predictable, but only after six positions in the sequence are observed by solving for

$$U = \begin{bmatrix} \mathbf{x}_1 & \mathbf{x}_2 & \mathbf{x}_3 & \mathbf{x}_4 & \mathbf{x}_5 \end{bmatrix} \begin{bmatrix} \mathbf{x}_0 & \mathbf{x}_1 & \mathbf{x}_2 & \mathbf{x}_3 & \mathbf{x}_4 \end{bmatrix}^{-1}. \tag{5}$$

### C.1 OPTIMAL PSEUDOINVERSE PREDICTOR

Following from equation 5, given the state observations $\{\mathbf{x}_0, \ldots, \mathbf{x}_i\}$, an optimal predictor for this problem computes $\widehat{\mathbf{x}}_{i+1} = \widehat{U}\mathbf{x}_i$, where

$$\widehat{U} = \begin{bmatrix} \mathbf{x}_1 & \ldots & \mathbf{x}_i \end{bmatrix} \begin{bmatrix} \mathbf{x}_0 & \ldots & \mathbf{x}_{i-1} \end{bmatrix}^{\dagger}, \tag{6}$$

and $X^{\dagger}$ denotes the Moore-Penrose pseudoinverse of $X$. Essentially, this baseline only makes non-zero errors on the first, second, third, fourth, fifth, and sixth entry in any sequence — it gets everything else perfectly correct.

### C.2 DATA GENERATION AND TRAINING

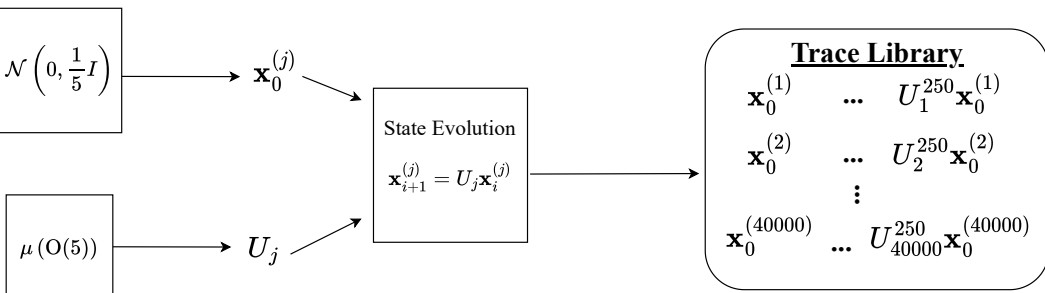

Figure 9: The generation of a train or test library of sequences.

**Generating a library of training sequences:** Depicted visually in Fig. 9, we first compile a training library by the following method:

1. Generate 40000 orthogonal matrices i.i.d. uniformly over all orthogonal matrices $U_1, \ldots, U_{40000} \overset{iid}{\sim} \mu\left(\mathrm{O}(5)\right)$, where $\mu\left(\mathrm{O}(5)\right)$ is the Haar measure over orthogonal matrices in $\mathbb{R}^{5 \times 5}$. (Mezzadri, 2006)

2. Generate 40000 i.i.d. initial states that will correspond to each training system $\mathbf{x}_0^{(1)}, \ldots, \mathbf{x}_0^{(40000)} \overset{iid}{\sim} \mathcal{N}\left(0, \frac{1}{5}I\right)$.

3. Roll out the states to get observation sequences that are each 251 entries long, and compile the sequences as our training library.

**Cutting and interleaving training sequences** To form a training trace, we interleave segments of observation sequences from the library into a context window of length 251, by this process:

1. Insert the start symbol at index 0.

2. Sample the maximum number of systems in the trace $N$ from a $\mathrm{Zipf}(1.5, 25)$ distribution depicted graphically[8] in Fig. 10a. This means that no more than 25 systems will ever appear in a training trace.

3. Choose $N$ of the 40,000 systems in the training library uniformly at random without replacement.

---

[8]The Zipf distribution was chosen for its ubiquity in nature (Bak, 1996) and natural language (Schutze & Manning, 1999), along with recent work pointing to its importance in modern neural networks (Michaud et al., 2023).

4. Randomly assign to each of the $N$ systems a pair of symbolic open and close labels for this training example.

5. Sample the number of cuts $C \sim \mathrm{Poisson}(2N)$ to be made in the trace. This means that there will be $C + 1$ trace segments in the trace.[9]

6. Place the $C$ cuts uniformly at random with replacement within the context window.

7. For each segment created by the cuts, in order, uniformly at random choose one of the $N$ systems with replacement.

8. At the cut at the beginning of the segment, the open label for this segment's system is inserted.[10]

9. For the system chosen, check if it has appeared in a previous segment of the trace. If not, insert the system's segment from the training trace library starting at index 0. If this system has appeared in a previous segment, insert the system's segment from the training trace library starting at the index that corresponds to the continuation of the previous segment for this system.

10. At one index before the next cut, insert the close label for this segment's system.

Note that within a single training example, segments of a particular system always start with the same open token and always end with its corresponding close token. These random assignments are redrawn at the beginning of the interleaving process for each training example; therefore, *the same system can have different symbolic open and close labels when it appears in different training examples*. See Fig. 2 for a diagram of an interleaved training example.

Given this randomized procedure for generating interleaved training examples, we can analyze the training distribution to better understand how frequently a model must recall a system, or sees many systems in a trace.

In Fig. 10, we show relevant distributions that are derived from the randomized interleaving procedure in this section. Figs. 10a and 10c show the $\mathrm{Zipf}(1.5, 25)$ distribution for the maximum number of systems per trace in black and the number of unique systems per trace in silver. The frequency of number of unique systems per trace follows closely the $\mathrm{Zipf}(1.5, 25)$ distribution for the smaller quantities of systems, but diminishes quicker for larger numbers of systems, due to the coupon-collecting phenomenon of picking the same system multiple times in a trace. The PMF for the number of cuts made in an interleaved trace is shown in Fig. 10b, while the CCDF for this quantity is given in Fig. 10d. Fig. 10e shows the frequency of how many times a system appears in a training trace. If the same system appears more than once, than the model must perform recall. Therefore, Fig. 10e gives an idea of how often the model must recall a system during training. Finally, Fig. 10f provides the frequency of the number of previously seen systems in the training trace that are candidates to be continued when a predictor is tasked to recall a system. The value zero on the x-axis of this figure means that the model is seeing a system for the first time in a training example and has no need to recall. Later, in Section E.1.2, we construct tests for the associative recall ability of the trained model for different numbers of candidate systems to be continued in the trace. Fig. 10f shows that for 19 candidate systems, the largest number of candidate systems that we tested on, the model has been presented with this situation less than $1\%$ of the time during training.

---

[9]On average, each training trace has $2N + 1$ segments to ensure that the trained model has seen ample interruptions and continuations of systems.

[10]Since the open and close labels occupy two indices in the context window, there are three special cases that can occur: (1) If two cuts are sampled to be on top of each other, then the first of the two cuts that were sampled is ignored; (2) If the two cuts are sampled to occupy adjacent indices, then only the close label for the system corresponding to first of the two cuts is inserted, effectively making that index meaningless as close labels are masked; (3) If the two cuts are sampled so that there is only one index between them, then the open label for the system corresponding to the first of the two cuts is inserted and is immediately followed by the close label for that system, effectively making both indices meaningless due to the masking of the labels. Note that the distributions shown in Fig. 10 do not account for these rare special cases.

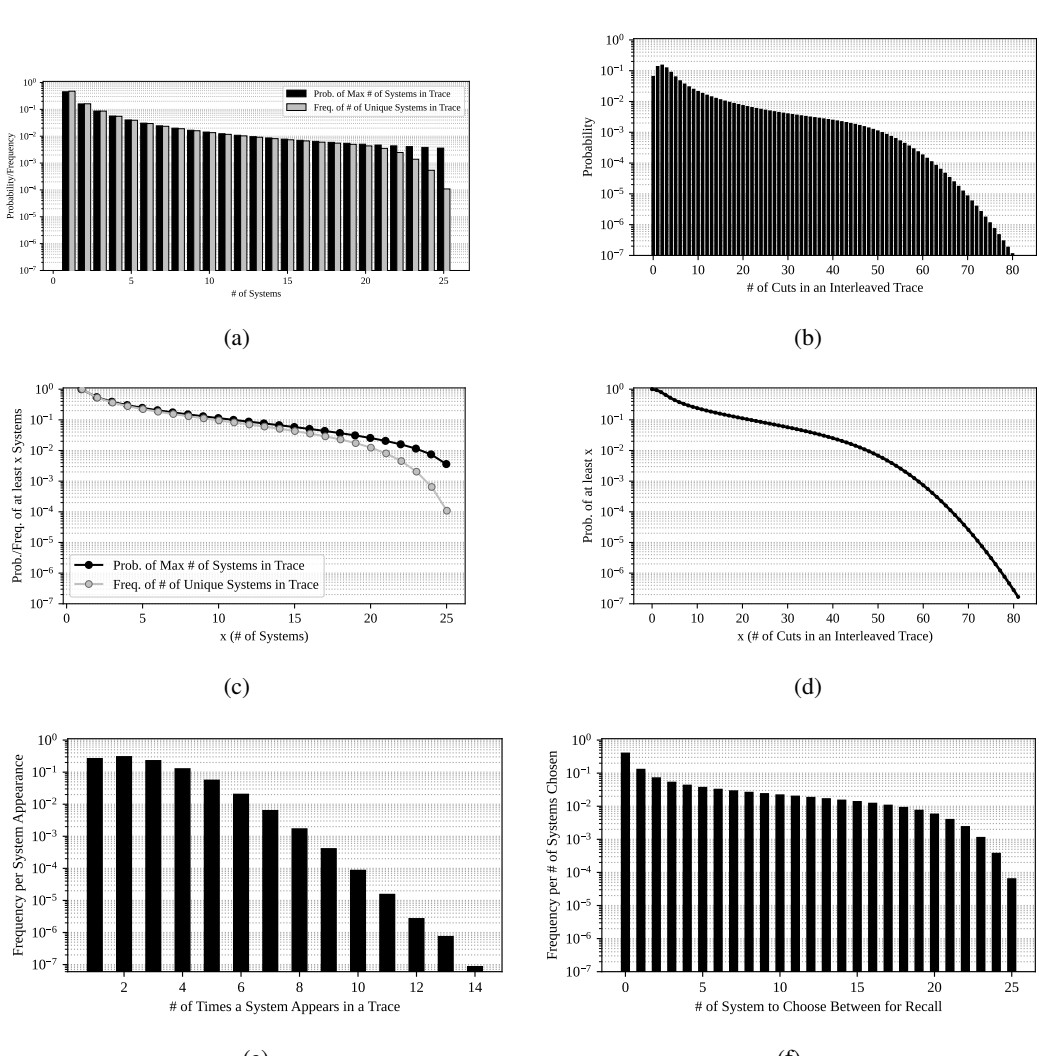

(a)

(b)

(c)

(d)

(e)

(f)

Figure 10: Distributions and complementary cumulative distribution functions (CCDFs) used in data generation — Fig. 10a is the $\mathrm{Zipf}(1.5, 25)$ distribution for the maximum number of systems per trace in black and the number of unique systems per trace for $1 \times 10^7$ traces in silver. Fig. 10c shows the CCDFs for these distributions. Fig. 10b shows the PMF and Fig. 10d shows the CCDF of the number of cuts per trace. Fig. 10e shows the frequency of the number of times a system will appear in the same trace per system appearance. Lastly, Fig. 10f shows the frequency of the number of previously seen systems a predictor must choose between to recall in a trace per system appearance. For example, if system 0 is chosen then system 1, then system 0, then the model must choose between two systems to recall.

**Input structure and embedding**    The input dimension of our models is 57. There are 50 dimensions for encoding paired symbolic open and close labels, a dimension for the start symbol, a dimension for the payload flag and 5 dimensions to hold the 5-dimensional observation vectors. The special symbols are one-hot encoded vectors; see Fig. 11 for an example of the open symbol. For the observation sequence between the SPLs, the 5-dimensional state vectors are inserted into the payload portion of the input vector, the payload flag is set to 1, and the rest of the input vector dimensions are zeroed out.

| 0 | $\sqrt{2}$ | 0 | $\cdots$ | 0 | 0 | 0 | 0  0  0  0  0 |
|---|---|---|---|---|---|---|---|
| Start Index | 0 Open | 0 Close | | 24 Open | 24 Close | Payload Flag | Payload |

Figure 11: The one-hot encoding of an open symbolic label. In this example, the system corresponding to this label is assigned to be "system 0."

The entire randomized procedure of generating interleaved training traces is depicted in Fig. 12 along with the structure of the inputs into the model.

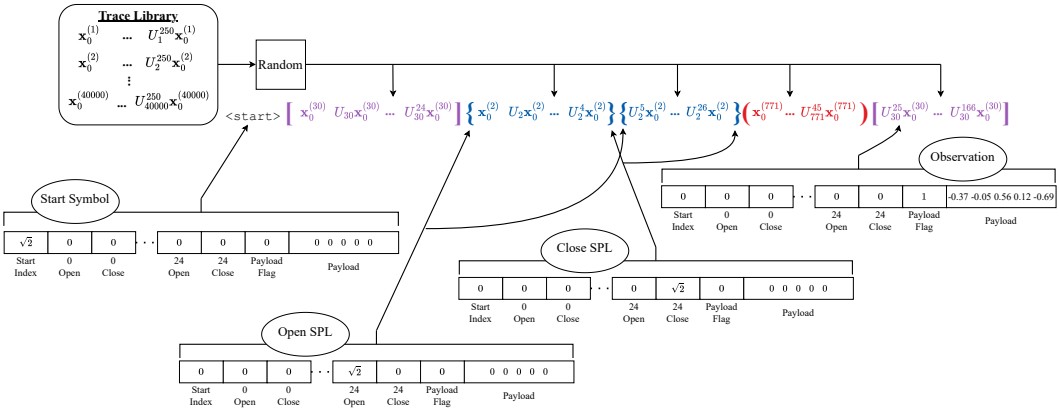

Figure 12: Generating a training example — Notice in this example the continuation from the first segment to the last (system $U_{30}$), and from the 2nd segment to the 3rd (system $U_2$). The "parentheses" (symbolic punctuating labels) are encoded as special tokens as shown.

**Model and embedding**    Building off of the codebase in (Du et al., 2023), which was influenced by (Garg et al., 2022), we train a 2.42M parameter GPT-2 style transformer to perform this task. Our model[11] has hidden dimension 128, 12 layers, and 8 heads. Our model's input embedding is $128 \times 57$ dimensional. The model's output layer is $5 \times 128$ to ensure that the model makes 5-dimensional predictions. The input and output layers are untied (Du et al., 2023).

**Training and Hyperparameters**    New interleaved training examples are generated for each training iteration and our GPT-2-style model was trained for next token prediction on these training traces.[12] The loss for all SPLs on the output were zeroed out. A model that successfully recalls the state of a system seen previously in its context should make predictions after the open token that perform as it would have if the relevant sequence had continued on without interruption.

Following the choice made in (Du et al., 2023), we trained our model with a weight decay of $1 \times 10^{-2}$. We used a batch size of 512, a learning rate of $\approx 1.58 \times 10^{-5}$, and trained on a single NVIDIA L40S

---

[11]These parameter counts and model dimensions are for our "medium" model. Three other models of different sizes were also trained, and their model dimensions are given table 2 in Appendix J.

[12]The model sees newly interleaved training examples at each iteration, but the training traces that are interleaved into the training example are drawn from the fixed training library of 40,000 sequences. Therefore, the model undergoes single-epoch training where the information within a training example might have appeared in many other training examples.

GPU with 48GB of RAM. A single training run takes around 5 days. We used the AdamW Optimizer (Loshchilov & Hutter, 2019) and trained using mean-squared error loss.

## D    FORMALIZING LABEL- VS. OBSERVATION-BASED RECALL

**Notation.**    Let $(y_0, y_1, y_2, \ldots)$ be the token stream given to the transformer. The tokens comprise of: (i) symbolic punctuating labels (SPLs) identifying systems, and (ii) continuous observations in $\mathbb{R}^5$. At index $t$, let $\mathcal{S}_t$ be the set of systems introduced so far, and let $m_t \in \mathcal{S}_t$ denote the system currently being queried. For any $m \in \mathcal{S}_t$, denote the observations of $m$ seen up to $t$ by

$$o_{m,0}, \ o_{m,1}, \ \ldots, \ o_{m,\mathrm{last}_t(m)} \in \mathbb{R}^5,$$

where $\mathrm{last}_t(m)$ indexes the most recent observed state of $m$ by time $t$.

**Pseudoinverse operator and observation compatible systems.**    Given $k \geq 1$ observations of system $m$ available by index $t$, define

$$\mathrm{PI}(m, t; k) \triangleq \begin{bmatrix} o_{m,1} & o_{m,2} & \cdots & o_{m,k} \end{bmatrix} \begin{bmatrix} o_{m,0} & o_{m,1} & \cdots & o_{m,k-1} \end{bmatrix}^{\dagger}. \tag{7}$$

In our noiseless setting with state dimension 5 and trace length at least 10, the right matrix in equation 7 has full row rank with probability 1, so $\mathrm{PI}(m, t; k)$ exactly recovers the true dynamics matrix $U_m$ (i.e., $\mathrm{PI}(m, t; k) = U_m$ whenever $k \geq 5$).

Let the set of observation-compatible systems $Q_t = \{m \in S_t \mid PI(m, r_t)y_h = y_\ell \ \forall \ell \in (r_t, t]\}$, where $r_t$ = position of the nearest open SPL preceding (or at) index $t$, and $h = \ell - 1$ if $\ell - 1 > r_t$ and $h = o_{m,last_t}$ otherwise.

**H1: Label-based recall.**    The model uses in-context learning to associate symbolic labels with systems and, upon a query at index $t$, deterministically recalls the system $m_t$ referenced by the most recent open parenthesis. The predictive distribution for the next observation is a distributed at

$$\alpha_t \triangleq \mathrm{PI}\big(m_t, t; k_{m_t, t}\big) \, o_{m_t, \mathrm{last}_t(m_t)}, \tag{8}$$

Under our noiseless orthogonal-system setting with sufficiently many past observations ($k_{m_t, t} \geq 5$), we have $\alpha_t = U_{m_t} o_{m_t, \mathrm{last}_t(m_t)}$ exactly.

**H2: Observation-based recall.**    The model ignores symbolic labels and infers the generating system only from observation consistency. Conditioned on $Q_t$, an observation-based predictor with a uniform prior over compatible systems yields

$$\widehat{y}_{t+1} = \frac{1}{|Q_t|} \sum_{m \in Q_t} \mathrm{PI}\big(m, t; k_{m, t}\big) \, y_t, \tag{9}$$

the average of one-step predictors from all systems consistent with the observations seen.

## E    EXTENDED ANALYSIS ON ICL FOR INTERLEAVED SYSTEMS

The toy problem, extensively described in Appendix C, has two qualitatively different capabilities that a model can acquire during training: in-context learning a linear system as it is seen (ICL), and recalling what has been in-context learned about a previously seen system (associative recall). Within the ICL ability, again, there are qualitatively two sub-abilities: ICL for the first system that is seen, and ICL for the subsequent new systems that are seen. The second sub-ability requires a model to learn to restart its in-context learning for a new system.

### E.1    TEST SETUP

We explain the our basic test setups to analyze the learning capabilities of models on the interleaved systems.

### E.1.1 Uninterleaved sequence test

To test the model's ICL ability for the first system that is seen (Section E.2.1), we generate 100 held-out systems and 1000 different held-out initial states[13] by the same method described in Section C.2 to form our testing library. We then evaluate the model on the uninterleaved traces from this testing library.

### E.1.2 Needle-in-a-haystack test

To evaluate the model's ability to restart ICL on a new system (Section E.2.2) and recall a previously seen system (Section E.3), we generate a series of structured "needle-in-a-haystack" test traces through interleaving the traces in the testing library generated in Section E.1.1. A single "needle-in-a-haystack" trace is generated by the following procedure:

1. Choose $N \in [1, 19]$ to be the number of distinct systems in a test trace.

2. For each of the $N$ systems, insert a segment of 10 state observations starting from index 0 from the testing library into the "needle-in-a-haystack" test trace. Each of these segments are individually punctuated with a unique open and close symbol pair. We call this portion of the trace the "haystack".

3. Append a query open symbol to the test trace that signifies which system in the haystack will be be continued. The segment that will be continued is called the "needle".

4. Append 10 state observations from the continuation of the system in the haystack corresponding to the query open symbol. This portion is called the "test segment".

See Fig. 13 for a diagram of a test trace for $N = 2$ systems in the haystack and system $U_1$ as the needle.

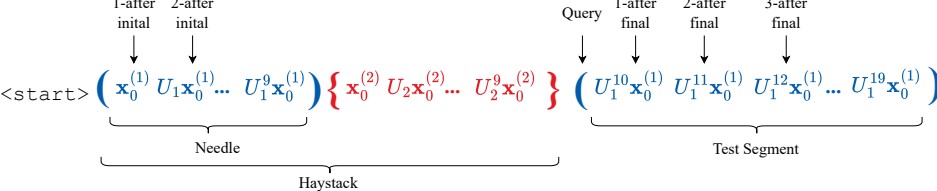

Figure 13: Needle-in-a-haystack test example. (A two system haystack.)

For the full "needle-in-a-haystack" test dataset, we would like to ensure that we test on the same systems for the different values of $N$, while having diverse systems in our dataset so that our results are statistically meaningful. To achieve this, we test on 50 "needle-in-a-haystack" trace configurations. A trace configuration is a specific ordering of systems from the testing library in the positions of the haystack. For the first needle-in-a-haystack trace configuration that we generate, we place a segment from "system 0" from the testing library into the first position in the haystack, and fill the rest of the haystack positions consecutively until the $N^{\text{th}}$ position is filled with a segment from "system $N - 1$". For the next trace configuration, the first position in the haystack is filled with a segment from "system 1" and the rest of the haystack positions are filled consecutively until the $N^{\text{th}}$ position is filled with a segment from "system $N$". This pattern continues until the last trace configuration. In our case, we tested on 50 trace configurations, meaning the haystack of the last trace configuration started with "system 49" and ended with "system $48 + N$". Each trace configuration is populated with 1000 different initial states for each system. For the results in the main paper, the test segment is a continuation of the segment in the first position of the haystack. For results where segments in other haystack positions are continued in the test segment see Section M in the Appendix.

### E.2 In-context learning system dynamics

We now present results on how well the transformer-based model can in-context learn a linear system's dynamics. Section E.2.1 shows the results for learning about the first system that is seen,

---

[13]We generate 1000 initial states for each system to narrow down the quartiles in the squared-error curves.

$$\begin{matrix} \{0 & \dots & N-1\} \\ \{1 & \dots & N\} \\ \vdots & \vdots & \vdots \\ \{49 & \dots & 48+N\} \end{matrix}$$

Figure 14: When testing on 50 needle-in-a-haystack trace configurations, the order of system indices from the testing library in a haystack of size $N$ for each needle-in-a-haystack test trace is given above. For each system, 1000 sequences are interleaved to build a testing dataset of shape $50 \times 1000 \times (12N+1) \times 5$. The shape of the second to last axis is due to the start token and haystack segments being 10 context indices long plus two indices for the symbolic open and close labels. The last axis is 5-dimensional since every system has 5-dimensional observations.

while Section E.2.2 shows the results for learning about subsequent systems that are seen and the ability to restart ICL.

### E.2.1  CAN A MODEL LEARN THE FIRST SYSTEM IN-CONTEXT? YES

We first confirm that our trained transformer model is able to in-context learn the first system seen in its context. We evaluate the model on the uninterleaved traces from the testing library specified in Section E.1.1. In Fig. 15a, we plot the median squared-error over these test traces vs the context index, and the color of each curve represents how far along the model is in training. The dotted line in this figure is the median squared-error of the pseudoinverse predictor in equation 6 over the same test traces. In Fig. 15b we plot the median squared-error over these test traces as training proceeds (measured by the number of training examples seen so far), the color of each solid curve represents the context index, the blue and green dotted horizontal lines are the optimal pseudoinverse predictor's median squared error for the specific early context indices.

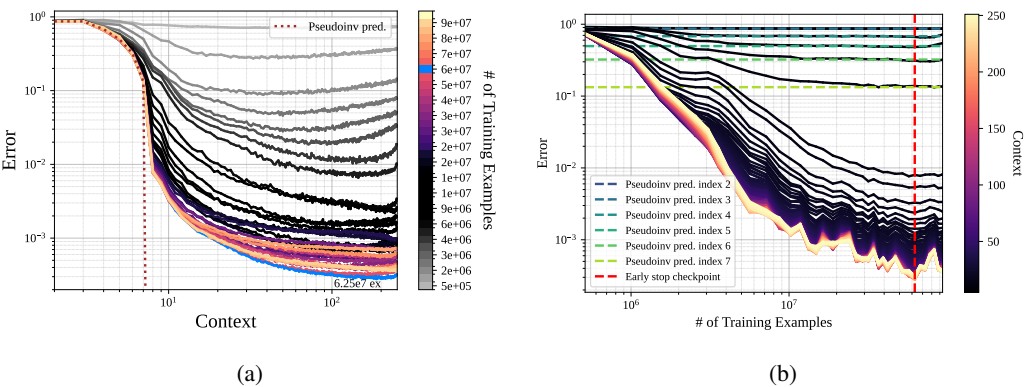

(a)

(b)

Figure 15: Performance on a long uninterleaved trace — 15a and 15b depict the in-context learning performance of the model on long uninterleaved test examples. The median squared-error is plotted against the context index in Fig. 15a, and against the number of training examples seen in Fig. 15b. The red line in Fig. 15b at $6.25 \times 10^7$ training examples, and the blue curve in Fig. 15a denote the checkpoint that we use for early stopping. in Fig. 15a the optimal pseudoinverse predictor is the brown dashed curve, while in Fig. 15b it is denoted by the blue and green horizontal lines for each of the early context indices. in Fig. 15b, notice that the model gradually learns to make optimal predictions as opposed to the sudden emergence of associative recall ability seen in Section E.3, since the prediction errors for early indices slowly converge towards the pseudoinverse predictor's performance.

Notice in Fig. 15a that the early model checkpoints saturate out and cannot continue to make better predictions with more context. Nevertheless, after seeing $6.25 \times 10^7$ training examples, the model's prediction error on indices 2 through 7 closely match those of the pseudoinverse baseline from equation 6. This is also seen in Fig. 15b where, as training proceeds, the dark curves representing

the model's prediction errors on early indices converge to the prediction error of the pseudoinverse predictor for the corresponding index given by the blue and green curves. The best performance of the model is at the end of the context window with a median squared-error of $\approx 2 \times 10^{-4}$. According to (Liu et al., 2024a), this is near the practical precision threshold for transformer models.

Notice in Fig. 15b the model is gradually learning to make better predictions as training continues. This ICL ability for the first system seen is the same ability that is studied by (Du et al., 2023; Sander et al., 2024) and using this evaluation metric, we see that sudden emergence is not present. Nonetheless, using the same evaluation metric, the ability to restart ICL for a new system (Section E.2.2) and to recall a previously seen system (Section E.3) both exhibit emergence.

Additionally, we notice in Fig. 15b that the squared-error bottoms out after $6.25 \times 10^{7}$ training examples, showing that the model suffers from overfitting late in training. Having seen this, we set our early stopping checkpoint at $6.25 \times 10^{7}$ training examples, as denoted by the red vertical line in Fig. 15b which corresponds to the blue curve in Fig. 15a.

In summary, the model is able to use context to make better predictions of state observations from held-out test systems. The model's ability gradually develops during training, as opposed to how associative recall develops suddenly later in training as seen in Section E.3.

### E.2.2 EMERGENCE OF THE ABILITY TO RESTART ICL ON NEW SYSTEMS

We now show the training dynamics for the ability to restart in-context learning for a new system that was not the first system seen in-context. We find that learning this restart ability is not gradual, as it was for learning the first system seen (Section E.2.1). Instead, we see that the model begins to transition from poor restart performance to good restart performance early in training as compared to when associative recall emerges in training which we will see in Section E.3. We study the model's performance on the haystack segments of "needle-in-a-haystack" test examples (see the diagram in Fig. 13).

In Fig. 16, the median squared-error vs the number of training examples seen is plotted for steps 1 through 8 into the first system segment in the haystack and the third system segment in the haystack. Specifically, in Fig. 16a we see that at the beginning of training the model has not learned to restart its predictions for the third system segment, as its median squared-error in segment 3 is well above its counterpart predictions in segment 1 for all steps into each segment except for step 1. This shows that early on in training, there is clearly substantial interference from earlier segments when the model tries to learn to predict the behavior for a new sequence (explicitly labeled as such) that it is seeing in the third position in the haystack. As training proceeds, we see that the median squared-error for each step in segment 3 converges to the value of its segment 1 counterpart. Fig. 16b shows that the model transitions towards restarting ICL correctly when training has processed $\approx 2 \times 10^{6}$ training examples.

In Fig. 17, we show the median squared-error of the model's predictions for up to 8 steps into each new segment at the early-stopping checkpoint of $6.25 \times 10^{7}$ training examples. This log-scale plot shows that even at the end of training, the model's ability to restart ICL for new systems slowly degrades when predicting indices 6, 7 and 8 as more new systems are presented in the context window. This is seen as the upward trend in the green and yellow curves in Fig. 17.

### E.3 EMERGENCE OF ASSOCIATIVE RECALL

We now study the prediction performance of the model when queried for recall. In particular, how it depends on the index into the test segment. Furthermore, we study how the ability to predict different indices in the test segment develops differently during training. Particularly, the ability to predict the first index into the test segment develops much later in training and much more abruptly than the ability to predict the subsequent indices into the test segment.

To uncover this, we test the model on the "needle-in-a-haystack" test traces described in Section E.2.1 with different values of $N$, for the number of systems in the haystack. Fig. 18 shows results for $N = 1, 2$, and 5. Results for more $N$ values are given in Appendix K. In Fig. 18 the solid curves marked with dots show the median squared-error of the model's predictions on the first, second, third, seventh, and eighth indices into the test segment. These curves are labelled as "after final" in the legend, because their indices are after the final query open symbol. To contrast, the dashed curves

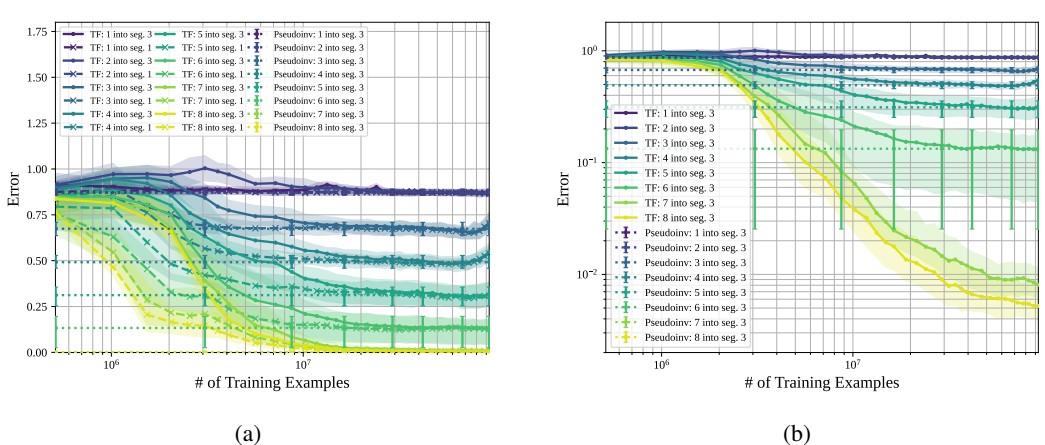

(a)                                                    (b)

Figure 16: Performance on new subsequent segments. 16a is the squared-error of predictions on steps 1 through 8 into the first and third system segments, where each segment is seen for the first time in context. 16b is the squared-error for steps 1 through 8 into the third system segments on log scale. The error bars show the $25^{\text{th}}$ and $75^{\text{th}}$ percentiles across trace configurations of the model's prediction error across the medians over the 1000 initial states in each trace configuration. The horizontal dotted lines are the median squared-error of the optimal pseudoinverse predictor.

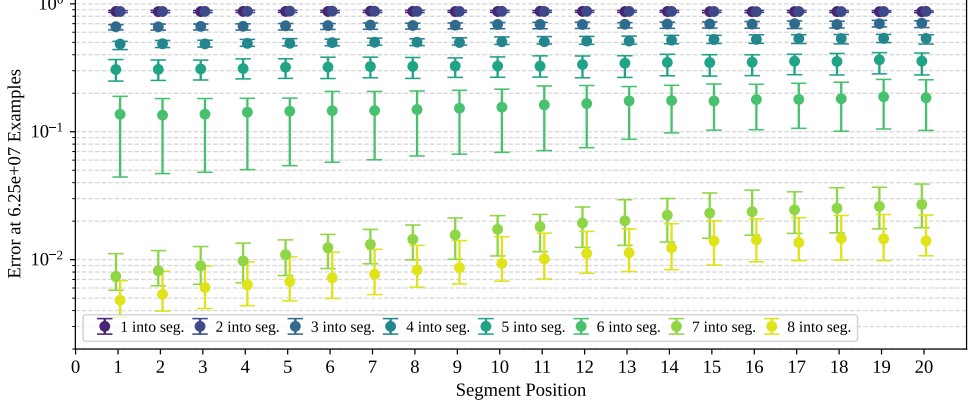

Figure 17: Restarting for a new system at the early-stopping checkpoint after seeing $6.25 \times 10^7$ training examples.

marked with crosses show the median squared-error of the model's predictions on the same indices into the first segment in the haystack. These curves are labelled as "after initial" in the legend, since their indices occur directly after the initial open symbol. These indices are depicted in Fig. 13.

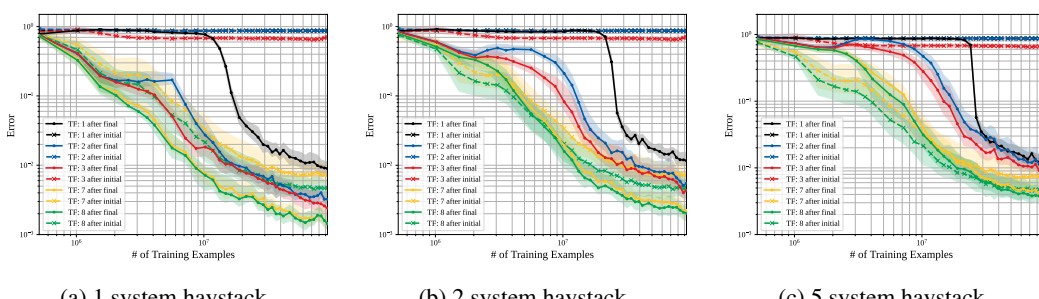

(a) 1 system haystack.      (b) 2 system haystack.      (c) 5 system haystack.

Figure 18: Training dynamics for recall — The $25^{th}$, $50^{th}$, and $75^{th}$ quartiles of the squared-error of the model's predictions vs the number of training examples seen during training so far are plotted on log-log plots for $N = 1$ in Fig. 18a, $N = 2$ in Fig. 18b, and $N = 5$ in Fig. 18c. All haystack segments are of length 10 (excluding delimiting tokens). The test set consisted of 1,000 "needle-in-a-haystack" traces from each of 50 systems. The dashed curves marked with crosses show the performance for indices 1, 2, 3, 7, and 8 steps after the initial open symbol, while the solid curves marked with dots show the performance for the same indices after the query open symbol. Notice in all the above figures that the solid black curve, showing the model's ability to recall the correct system from just seeing its corresponding open symbol, very sharply improves after $10^7$ training examples.

To perform perfectly on the associative recall task, the model must implicitly learn and remember the correct $5 \times 5$ orthogonal matrix corresponding to a particular system. In all subfigures in Fig. 18, the solid curves largely decrease as training proceeds.[14] This shows that training is improving the model's ability to recall. More specifically, the solid black curve for the pure recall task of predicting the first observation in the test segment shows emergence-style behavior: first a steady high squared-loss until it begins to drop at some point in training. Interestingly, this ability begins to emerge *earlier* ($\approx 1.5 \times 10^7$ training examples seen) for the simple case of $N = 1$ (Fig. 18a), than for case when $N = 5$ where the transition happens closer to $2.5 \times 10^7$ (Fig. 18c). One system in the haystack is an arguably-trivial recall test, since the model must recall the only system that it has seen so far. While recall performance for $N = 5$ is showing seemingly no improvement in Fig. 18c, the recall for $N = 1$ in Fig. 18a has gotten substantially better. The whole time, we are also seeing improvements in recognizing the underlying state evolution of the time-series as evidenced by better progress in the other blue, red, yellow, and green solid curves — all of which involve predicting the correct orthogonal transformation of the observed state.

---

[14]For the solid blue and red curves showing the model's recall ability on the second and third indices after the query symbol, we notice double-descent behavior that is more pronounced for larger haystacks.

# F EXTENDED OOD EXPERIMENTS

## F.1 FIGURES FOR EXPERIMENT 1: MISDIRECTION TOWARDS THE INCORRECT SEQUENCE IN THE HAYSTACK

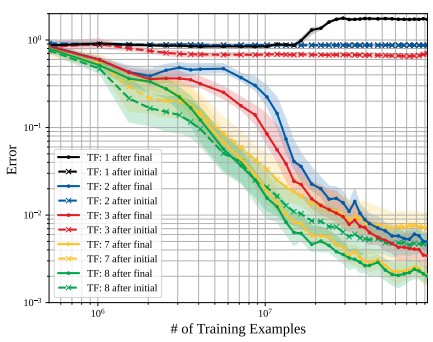

(a) 2 systems in the haystack.

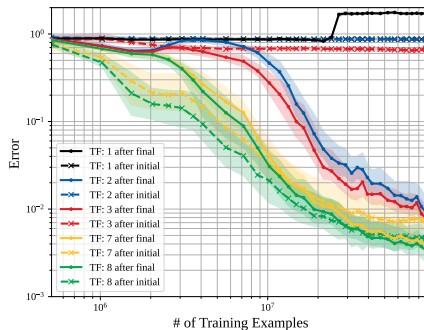

(b) 5 systems in the haystack.

Figure 19: Misdirection towards incorrect sequence — The median squared-error of the model's predictions on the test segment after observing the incorrect query open label vs the number of training examples seen so far. The solid black curve sharply increases in these figures where it would have sharply decreased for a normal test trace as seen in Fig. 18, suggesting H1 is true for predicting the first index of the test segment. In contrast, we find that the model ignores the misdirection when predicting two or more indices into the test segment, since the solid blue, red, yellow, and green curves are almost identical to the corresponding curve in Fig. 18. This suggests that the model is using an observation-based approach (H2-style) when predicting these indices.

## F.2 FIGURES FOR EXPERIMENT 2: SYNCHRONIZING "ROTATIONS" IN THE HAYSTACK

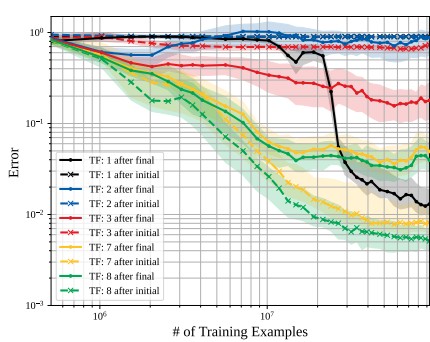

(a) 2 systems in the haystack.

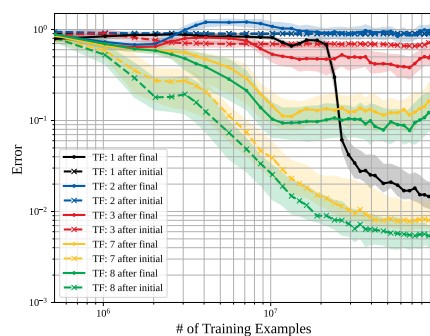

(b) 5 systems in the haystack.

Figure 20: Synchronizing rotations — The median squared-error of the model's predictions on the test segment when the first index into the test segment is a valid continuation for all haystack segments vs how many training examples have been processed during training. Synchronizing the rotations of the haystack segments means that after the first observation in the test segment, a predictor cannot determine which system is being continued. The solid black curve still shows emergence at the same point in training as it did in Fig. 18, supporting the validity of H1 for predicting the first index into the test segment. Otherwise, the model performance is significantly degraded for the rest of the indices into the test segment. For example, the solid blue curve in these plots for prediction errors on the second index into the test segment is near $1.0$ in Fig. 20a where it is near $1.0 \times 10^{-2}$ in Fig. 18b. This indicates that the model is unable to use the symbolic label well enough to make accurate predictions.

### F.3 EXPERIMENT 3: MISDIRECTION TOWARDS AN UNSEEN SEQUENCE

To further study whether the model uses the symbolic labels at all to continue its predictions on a recalled sequence, we misdirect the model to restart ICL for a sequence that has not appeared in the haystack so far. Fig. 21 illustrates how we swap out the query open symbol of a normal "needle-in-a-haystack" test trace with an open symbol that does not correspond to any of the systems in the haystack. Given the results from the previous two sections, we expect the model to predict zero for the first index as that is optimal for restarting a new sequence, but we expect the model to make accurate predictions on the subsequent indices that are continuing a segment that is in the haystack. This is almost what happens, but late in training, the model suddenly transitions to (at least partially) restarting ICL on a new sequence.

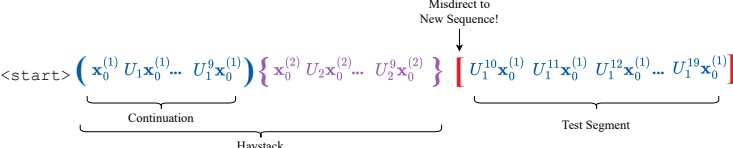

Figure 21: Misdirecting the model with an unseen symbolic label indicating a new sequence.

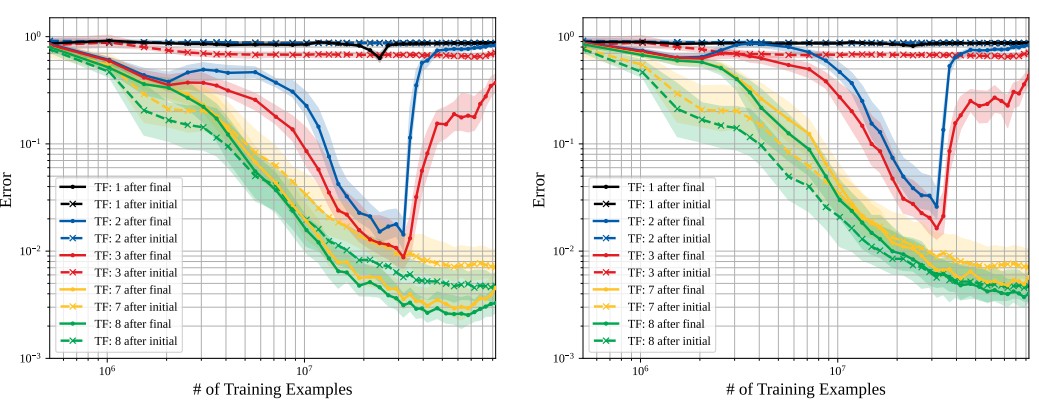

(a) 2 systems in the haystack.

(b) 5 systems in the haystack.

Figure 22: Misdirection towards an unseen system — The median squared-error of the model's predictions on the test segment when the query open symbol does not correspond to any haystack systems vs the number of training examples seen so far. The solid blue and red curves match their counterparts in Fig. 18 until $\approx 3 \times 10^7$ training examples, at which point they sharply increase. This means the model continues predicting the existing system in early training then suddenly starts treating it more like an unseen system late in training. We note that this transition happens shortly after the emergence of associative recall, suggesting the model has learned that unseen labels also require ICL to be restarted.

**Misdirecting with an unseen symbolic label highlights a new abrupt phase transition in model behavior late in training from disregarding the symbolic label and continuing to predict the observed test segment to restarting ICL for a new system.** Fig. 22 shows the median squared-error of the model predictions after being presented with a query open symbol that does not correspond to any system in the haystack while the test segment is actually a continuation of a haystack segment vs how far along the model is in training. In the same figure, the first after query predictions match the expected behavior of restarting predictions as seen by the solid black curve being a horizontal line near 1. For the rest of the indices, the model ignores the symbolic label through the initial stages of training and continues to correctly predict the test segment, since the solid curves in Fig. 22 match their counterparts in Fig. 18 up to around $3 \times 10^7$ training examples. Once associative recall fully emerges later in training, and the model learns how to use the symbolic labels, the model transitions to treating a continuation of the old sequence as a brand new sequence corresponding to the new label, since the blue and red solid curves abruptly increase after $3 \times 10^7$ training examples in Fig. 22.

This shows that the model predictions for two or more indices into the test segment *are* affected by the query open symbol, although Section 3.2.2 showed that the model is unable to use it to make accurate predictions for recall on those indices.

### F.4 EXPERIMENT 4: MISDIRECTION TOWARDS A SEEN SEQUENCE IN THE HAYSTACK

The first three out-of-distribution experiments studied the model's mechanisms for performing associative recall, but this section will study the mechanism for restarting its prediction on a new system. Again, we devise a misdirection experiment. This time we provide a query open symbol that points toward the "needle" in the haystack, but the test segment is a sequence from a system that is not in the haystack (Fig. 23).

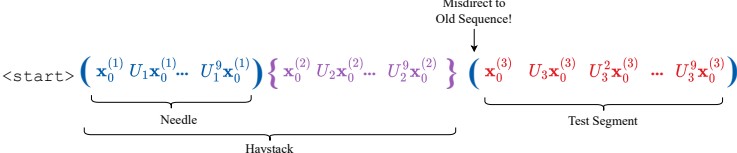

Figure 23: Misdirecting the model with a previously seen symbolic label.

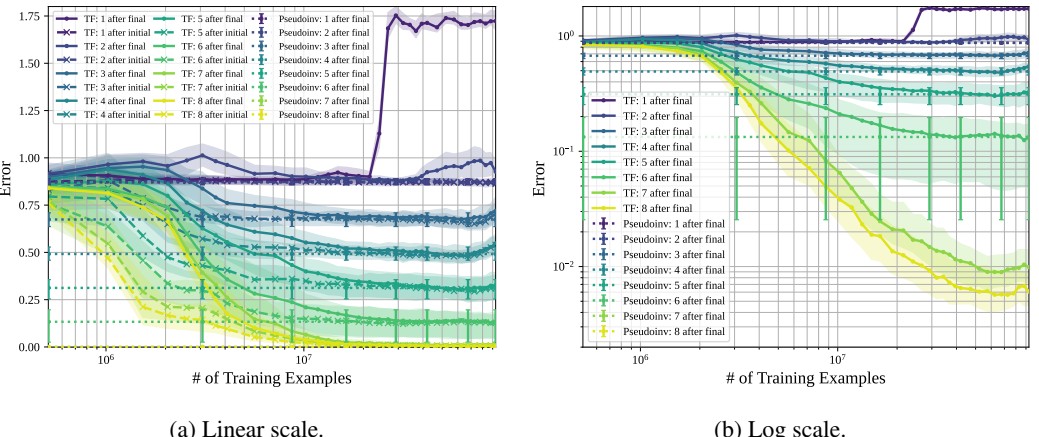

(a) Linear scale.                    (b) Log scale.

Figure 24: Misdirection towards a seen system — The median squared-error of the model's predictions on the third segment that is an unseen sequence while the query open symbol corresponds to the first segment vs the number of training examples seen so far. The solid curves other than the 1-after query curve match the solid curves in Fig. 16. This supports the hypothesis that the model uses the state observations to continue its prediction on a newly seen segment, rather than the open symbolic label.

**Misdirecting with a previously seen symbolic label does not stop the model from restarting its predictions on the new sequence for two or more indices into the test segment.** In Fig. 24, the median squared-error on this misdirection experiment is plotted against the number of training examples seen so far during training. We find that when given the correct unseen symbolic label (Fig. 16a), and when shown a symbolic label misdirecting to a sequence that has already been observed in context (Fig. 24), the model performs equivalently on predicting two or more indices into the newly seen segment, as the solid curves match except for the 1-after query curve that is using the symbolic open label. The model recognizes that the sequence it is seeing does not correspond to a sequence it has already seen. This supports a hypothesis that that model does not use the symbolic labels to restart ICL on a new system. Interestingly, misdirecting the model with a previously unseen symbolic label indicating a new system (Section F.3) shows that these unseen symbolic labels do affect the model's predictions in the test segment. This evidence shows that the mechanism for restarting ICL may not be purely label-based nor purely observation-based.

## G    EDGE PRUNING CIRCUIT VISUALIZATION

Section 3.3 describes how optimizing over continuous masks on disentangled transformer nodes leads to a sparse computation graph after the continuous masks are quantized to 0 or 1. To perform this quantization, the quantization threshold must be set. We use binary search to find the smallest threshold (within 1e-5 precision) such that the pruned model's edge sparsity is close to the target edge sparsity of 0.98. Importantly, this does not force a viable path from the start and end of the residual stream, it only captures significant contributions over edges with respect to the loss. In Fig. 25, the pruned circuit is visually presented in a directed computation graph. Nodes in this computation graph are specific QKV matrices of attention heads, entire attention heads, MLPs at a specific layer, or the output of the residual stream. The output of the residual stream is labeled as resid_post, MLPs are labeled as m{layer_num}, attention heads are labeled as a{layer_num}.h{head_num}, and .q, .k, and .v stand for QKV matrices respectively.

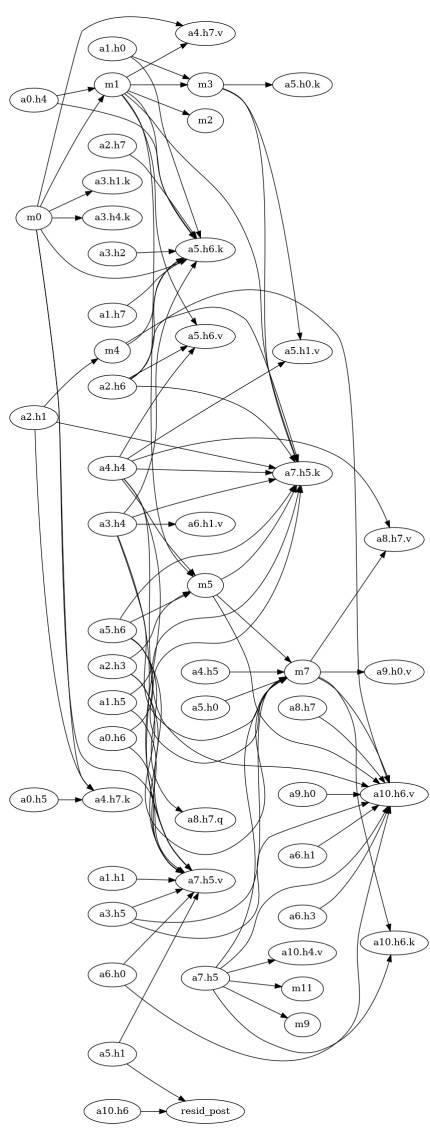

Figure 25: 1-after query open symbol circuit in the orthogonal model. The output of the residual stream is labeled as resid_post, MLPs are labeled as m{layer_num}, attention heads are labeled as a{layer_num}.h{head_num}, and .q, .k, and .v stand for QKV matrices respectively.

# H    NLP PROMPTS

In this section, we provide sample prompts for the NLP experiments from Section 4. We use 100 English prompts from the IPA transliterate benchmark task in BIG-bench (bench authors, 2023), and pass these English prompts through Google Translate to obtain their Spanish translations.

Below are examples of three different task instances for English to Spanish for the in-weights associative recall task.

```
"input":  Spanish:  Inició el ascenso del Imperio Británico en la
India.  English:
"target":  It started the rise of the British Empire in India.
"input":  Spanish:  Hicieron acusaciones sobre la plataforma.
English:
"target":  They made accusations about the platform.
"input":  Spanish:  Che fue otro dictador que ascendió al poder
mediante un levantamiento militar.  English:
"target":  Che was another dictator who rose to power by military
uprising.
```

Below are examples of three different task instances for English to Spanish for the in-context associative recall task. Notice that they are the same as the in-weights associative recall examples, except the semantically meaningful labels "English" and "Spanish" are swapped for the labels "X" and "Y".

```
"input":  X: Inició el ascenso del Imperio Británico en la India.
Y:
"target":  It started the rise of the British Empire in India.
"input":  X: Hicieron acusaciones sobre la plataforma.  Y:
"target":  They made accusations about the platform.
"input":  X: Che fue otro dictador que ascendió al poder mediante
un levantamiento militar.  Y:
"target":  Che was another dictator who rose to power by military
uprising.
```

For the counterpart to the misdirecting-to-an-unseen-token task, to adapt this task to a natural language setup, we keep our translation task with two few-shot examples exactly the same as in Section H, but alter the final testing prompt to not use the 'Y:' label tokens and to instead use a previously unseen 'Z:' token.

As such, at inference time, the full first token prompt functionally looks like this.

```
X: Inició el ascenso del Imperio Británico en la India.
Y: It started the rise of the British Empire in India.
X: Hicieron acusaciones sobre la plataforma.
Y: They made accusations about the platform.
X: Che fue otro dictador que ascendió al poder mediante un
levantamiento militar.
Z:
```

The above is an example for misdirection-to-an-unseen-token. If it was a normal case, the final "Z:" would be "Y:".

# I    EXPLORING PRETRAINING LOSS

In (Du et al., 2025), it was shown that many emergent abilities emerge for models of different sizes once a model's pretraining loss performance on a broad corpus of held-out data in the style of their pretraining data reaches a certain threshold. We want to see if this holds true in our toy setting of interleaved time-series prediction.

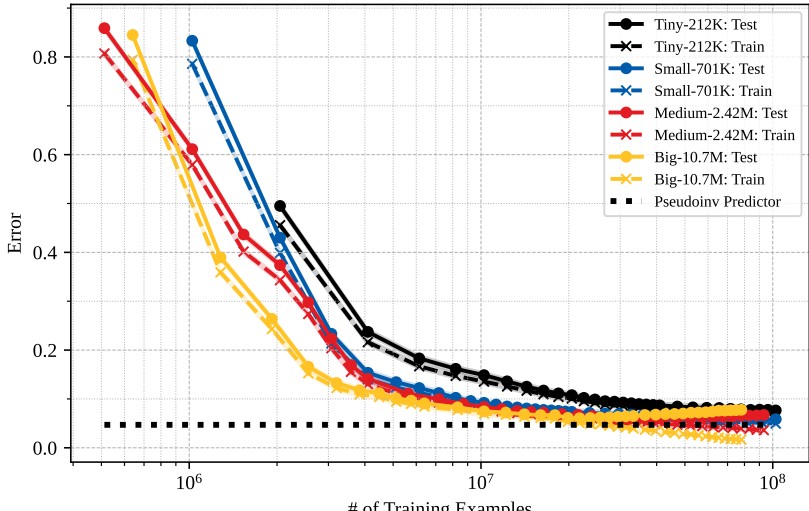

Figure 26: Pretraining loss — The squared-error of each transformer model's predictions on traces interleaved in the style of the training data averaged over each time step of the trace. For both the train and test data, averaging was done over 40,000 different interleaved traces, each of length 251. The horizontal black dotted line is the averaged squared-error of a predictor that computes an estimate of the underlying system dynamics by using the Moore-Penrose pseudoinverse of the observed data.

We evaluate four different model sizes (see Table 2 in Appendix J for model parameter details) and show how their pretraining loss dynamics differ throughout training. In Fig. 26, we see the performance of model training checkpoints on data that is in the style of what the model saw during training as specified in Section C.2. The dotted curves are the models' performance on freshly drawn interleavings of traces from a held-out test library, while the crossed curves are on freshly drawn interleavings of traces from the training library.

In Fig. 26, it is clear that larger models decrease their pretraining loss earlier in training than smaller models. Furthermore, classic overfitting behavior is evident, especially in the larger models. We see that the pretraining error on the held-out dataset deteriorates late in training, while the error on the training data continues to decrease. For the "big" model, its error on the training data even goes lower than the fundamental lower bound imposed by the error of the perfect pseudoinverse predictor.

### I.1 TRAINING DYNAMICS WITH RESPECT TO THE PRETRAINING LOSS

Taking inspiration from (Du et al., 2025), in Fig. 27 we look at when the ability to resume a recalled sequence, the ability to continue predicting a recalled sequence, and the ability to restart ICL on a previously unseen system emerge with respect to the pretraining loss as opposed to the number of training examples. This is to see if these abilities emerge at a specific pretraining loss threshold that is independent of the model size. This pretraining loss corresponds to the held-out loss specified in Section I. The red vertical line in the plots is the pretraining loss achieved by the pseudoinverse predictor specified in equation 6 averaged over all context indices. The blue horizontal line in Figs. 27e and 27f is the median error of the pseudoinverse predictor at the specific index corresponding to the curve plotted.

Although figures in Appendices J.1 and K show that larger models develop these emergent abilities before smaller models with respect to the number of training examples, in Fig. 27b we see that all the models exhibit the "1-after query" phase transition at a pretraining loss of around $6.5 \times 10^{-2}$. Furthermore, in Fig. 27d, the "2-after query" phase transition occurs for all models begin at a pretraining loss of a bit more than $1 \times 10^{-1}$. In Fig. 27a the different models are again tightly linked, except for the "tiny" model whose emergence for "1-after query" occurs at a pretraining loss that is around 0.02 larger than the rest. Overall, these results seems to qualitatively match the conclusions in (Du et al., 2025) that pretraining loss is a good indicator for when emergent abilities emerge.

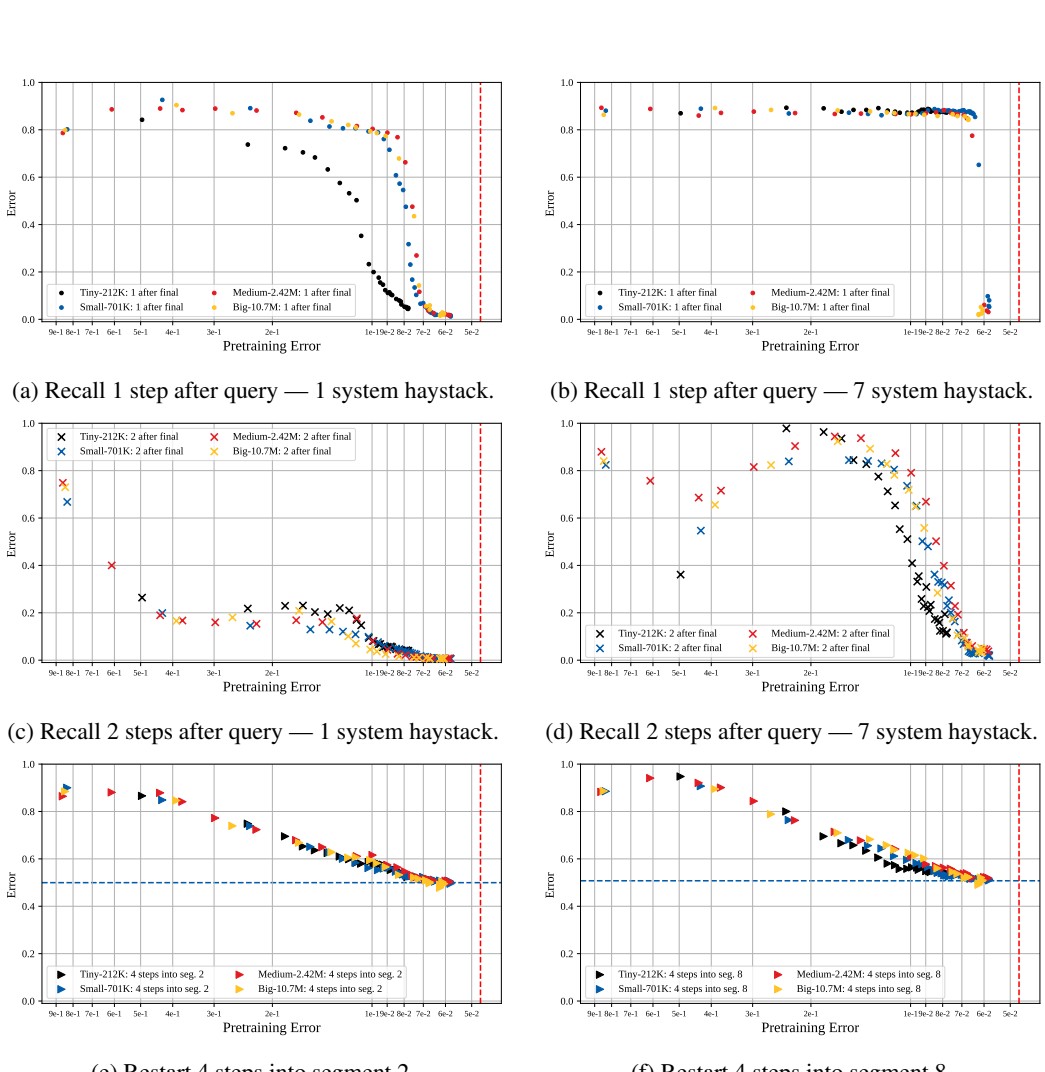

(a) Recall 1 step after query — 1 system haystack.

(b) Recall 1 step after query — 7 system haystack.

(c) Recall 2 steps after query — 1 system haystack.

(d) Recall 2 steps after query — 7 system haystack.

(e) Restart 4 steps into segment 2.

(f) Restart 4 steps into segment 8.

Figure 27: Recall and restart performance vs pretraining loss on held-out data. The red vertical line is the fundamental lower bound pretraining loss achieved by the pseudoinverse predictor. The blue horizontal line in Figs. 27e and 27f is the median error of the pseudoinverse predictor at the specific index plotted in each figure.

# J  THE EFFECTS OF MODEL SIZE

| Model Name | $n_{params}$ | $n_{layers}$ | $d_{model}$ | $n_{heads}$ | $d_{head}$ | Learning Rate | Batch Size |
|---|---|---|---|---|---|---|---|
| Orthogonal Tiny | 212K | 3 | 72 | 6 | 12 | $1.7 \times 10^{-4}$ | 2048 |
| Orthogonal Small | 701K | 6 | 96 | 6 | 16 | $4.5 \times 10^{-5}$ | 1024 |
| Orthogonal Medium | 2.42M | 12 | 128 | 8 | 16 | $1.6 \times 10^{-5}$ | 512 |
| Orthogonal Big | 10.7M | 24 | 192 | 12 | 16 | $1.5 \times 10^{-5}$ | 640 |
| Identity Tiny | 212K | 3 | 72 | 6 | 12 | $6.3 \times 10^{-5}$ | 8192 |
| Identity Small | 701K | 6 | 96 | 6 | 16 | $3.2 \times 10^{-5}$ | 4096 |
| Identity Medium | 2.42M | 12 | 128 | 8 | 16 | $1.6 \times 10^{-5}$ | 1024 |
| Identity Big | 10.7M | 24 | 192 | 12 | 16 | $1.3 \times 10^{-5}$ | 512 |

Table 2: Model size and training hyperparameters.

In order to test the effect of model size on our emergence results, we trained models across 4 different model sizes as shown in Table 2. We originally tuned the learning rate for the medium model with a batch size of 512 on a single GPU. Following the model scaling that was done[15] in (Brown et al., 2020b), when decreasing the size of our model from "medium" to "small" we halved the number of layers, multiplied the model dimension by $0.75$, maintained the same head dimension, and doubled the learning rate. To go from "small" to "tiny", we used the same process except we chose the head dimension to be 12 to maintain an integer value for the number of heads. If we would have maintained the head dimension of 16, the tiny model would have had $4.5$ heads. For this reason, 12 was chosen as the head dimension since it is the largest integer less than 16 that leads to an integer when dividing 72. To go from "medium" to "big", we doubled the number of layers, multiplied the model dimension by $1.5$, maintained the same head dimension, and multiplied the learning rate by $\frac{5}{6}$. This scaling was used for the identity models and batch size was not taken into account. It was later brought to our attention that the learning rate should also scale with the batch size. For our later orthogonal runs, we additionally adopted the square-root learning-rate scaling as indicated in (Li et al., 2024). Specifically, we took the learning rate we had scaled by model size and multiplied it by $\sqrt{\frac{batch\_size}{512}}$. However, we have not verified if this scaling is the best way to proceed with our training and further testing is still required. Furthermore, we trained the identity models on one Nvidia GH200 GPU with 80GB of RAM whereas we trained on one L40S GPU for the orthogonal "tiny", "small", and "medium" models and we trained on two L40S GPUs for the orthogonal "big" model. One L40S GPU has 48GB of RAM. This is the reason for the differing batch sizes across different model types.

## J.1  WHEN DOES THE ASSOCIATIVE RECALL ABILITY EMERGE FOR PREDICTING THE FIRST TEST SEGMENT OBSERVATION?

Evidence that larger models see earlier emergence is shown in Figs. 28, 29 and 30, where the number of training examples until the phase transition is shown for each model size for the identity and orthogonal systems. This was done by recording the checkpoint before and after the mean-squared error for the pure recall task dropped below the cutoff values of $0.4$ for the identity systems and $0.5$ for the orthogonal systems. These cutoff values were chosen by visual inspection.

---

[15]Alternatively, one could follow the model scaling done in (Groeneveld et al., 2024). There, they also halve the number of layers when decreasing the model size, but decide to halve the model dimension and number of attention heads as well.

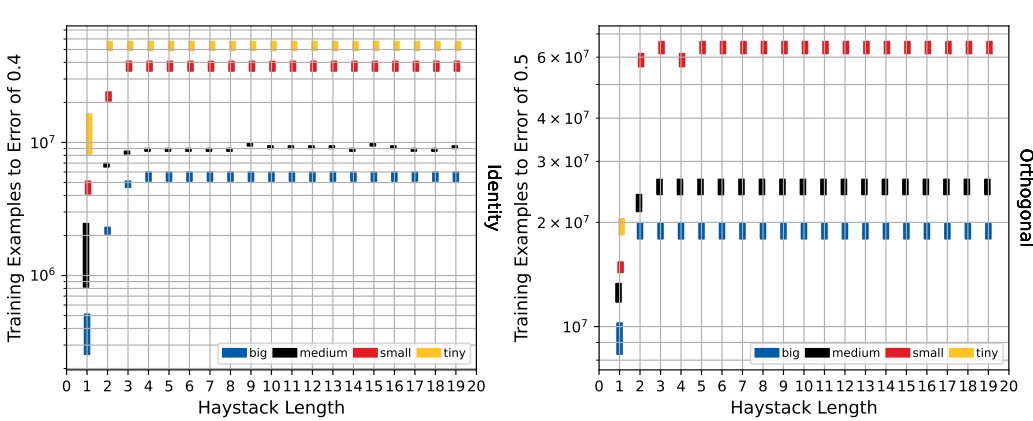

Figure 28: Emergence of associative recall on different haystack lengths across model sizes — The plot above shows that associative recall emerges much earlier for larger model sizes both when trained on identity systems as well as when trained on orthogonal systems. We also see that associative recall emerges earlier when there is only one system in the haystack but levels out as the model learns to generalize associative recall regardless of the haystack size.

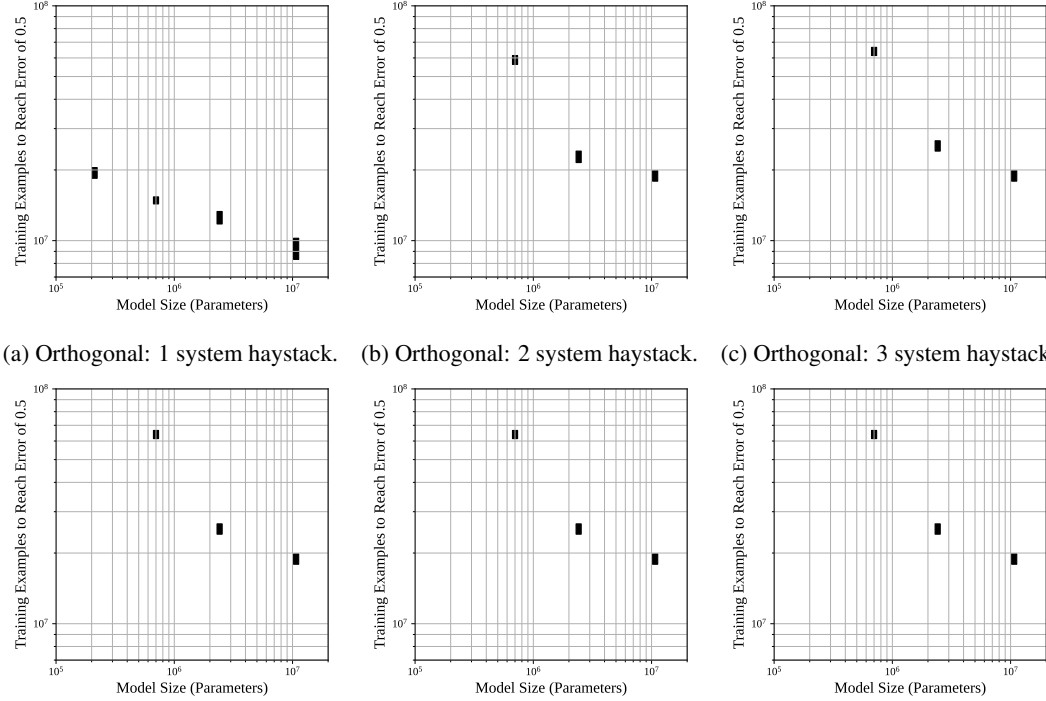

(a) Orthogonal: 1 system haystack.  (b) Orthogonal: 2 system haystack.  (c) Orthogonal: 3 system haystack.

(d) Orthogonal: 17 system haystack.  (e) Orthogonal: 18 system haystack.  (f) Orthogonal: 19 system haystack.

Figure 29: Emergence of associative recall in varying model sizes across haystack lengths for Orthogonal systems

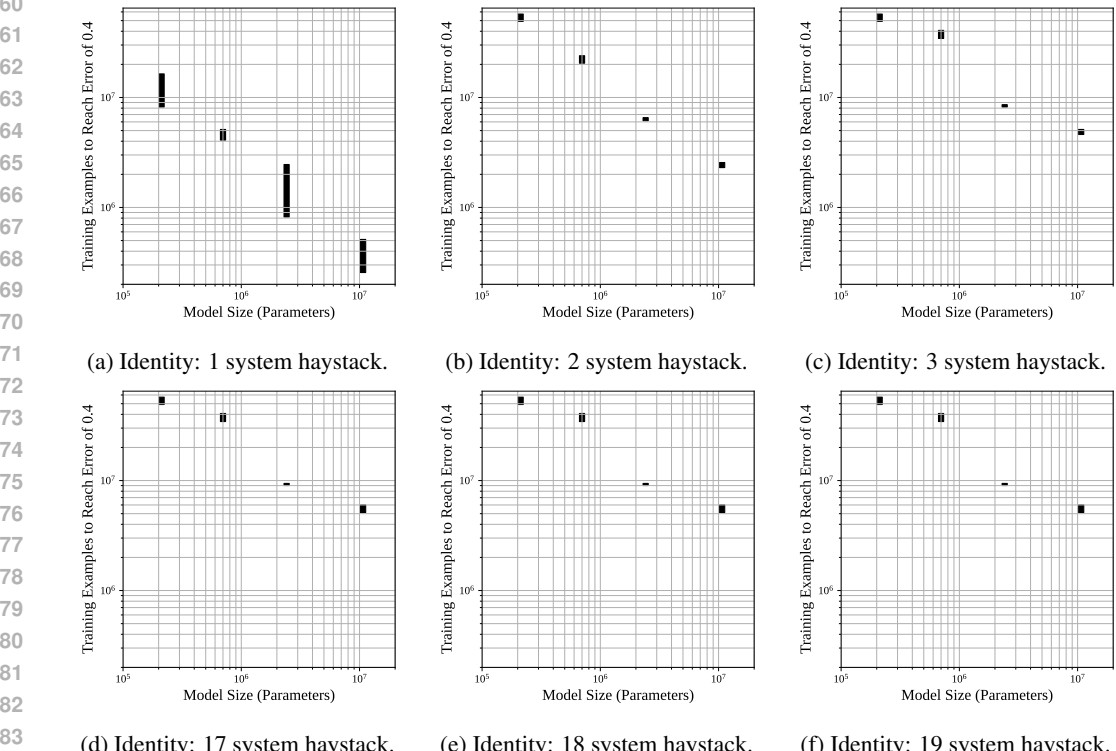

(a) Identity: 1 system haystack.  (b) Identity: 2 system haystack.  (c) Identity: 3 system haystack.

(d) Identity: 17 system haystack.  (e) Identity: 18 system haystack.  (f) Identity: 19 system haystack.

Figure 30: Emergence of associative recall in varying model sizes across haystack lengths for Identity systems

## K MORE TRAINING DYNAMICS PLOTS

Following the summary results from above, we provide full training dynamics plots for our different model sizes and show the results when tested on haystack lengths of 1, 2, 3, 17, 18 and 19. For your convenience here are the different figures: Orthogonal tiny (linear scale) Fig. 31, Orthogonal tiny (log scale) Fig. 32, Orthogonal small (linear scale) Fig. 33, Orthogonal small (log scale) Fig. 34, Orthogonal medium (linear scale) Fig. 35, Orthogonal medium (log scale) Fig. 36, Orthogonal big (linear scale) Fig. 37, Orthogonal big (log scale) Fig. 38, Identity tiny (linear scale) Fig. 39, Identity tiny (log scale) Fig. 40, Identity small (linear scale) Fig. 41, Identity small (log scale) Fig. 42, Identity medium (linear scale) Fig. 43, Identity medium (log scale) Fig. 44, Identity big (linear scale) Fig. 45, Identity big (log scale) Fig. 46

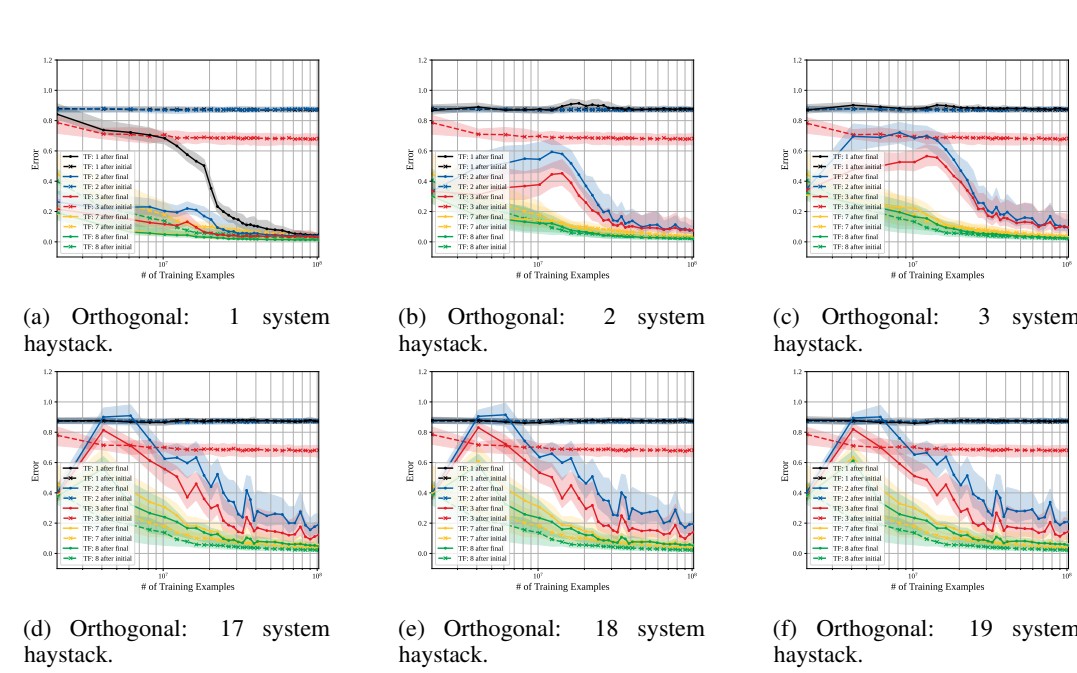

(a) Orthogonal: 1 system haystack.

(b) Orthogonal: 2 system haystack.

(c) Orthogonal: 3 system haystack.

(d) Orthogonal: 17 system haystack.

(e) Orthogonal: 18 system haystack.

(f) Orthogonal: 19 system haystack.

Figure 31: Performance of tiny orthogonal model (212K params) across training — linear-scale.

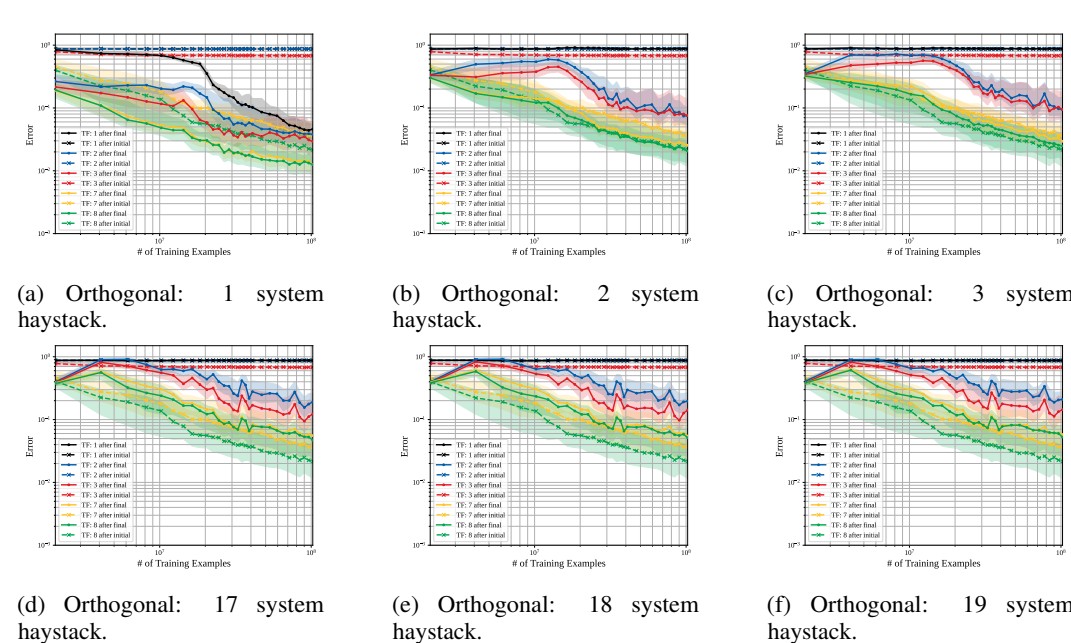

(a) Orthogonal: 1 system haystack.

(b) Orthogonal: 2 system haystack.

(c) Orthogonal: 3 system haystack.

(d) Orthogonal: 17 system haystack.

(e) Orthogonal: 18 system haystack.

(f) Orthogonal: 19 system haystack.

Figure 32: Performance of tiny orthogonal model (212K params) across training — log-scale.

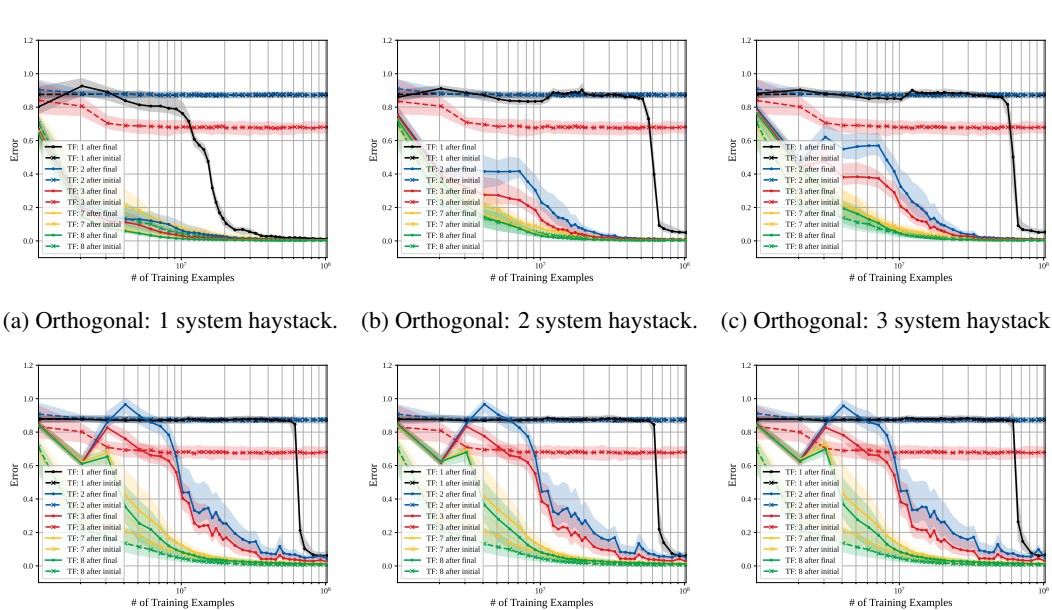

(a) Orthogonal: 1 system haystack. (b) Orthogonal: 2 system haystack. (c) Orthogonal: 3 system haystack.

(d) Orthogonal: 17 system haystack. (e) Orthogonal: 18 system haystack. (f) Orthogonal: 19 system haystack.

Figure 33: Performance of small orthogonal model (701K params) across training — linear-scale.

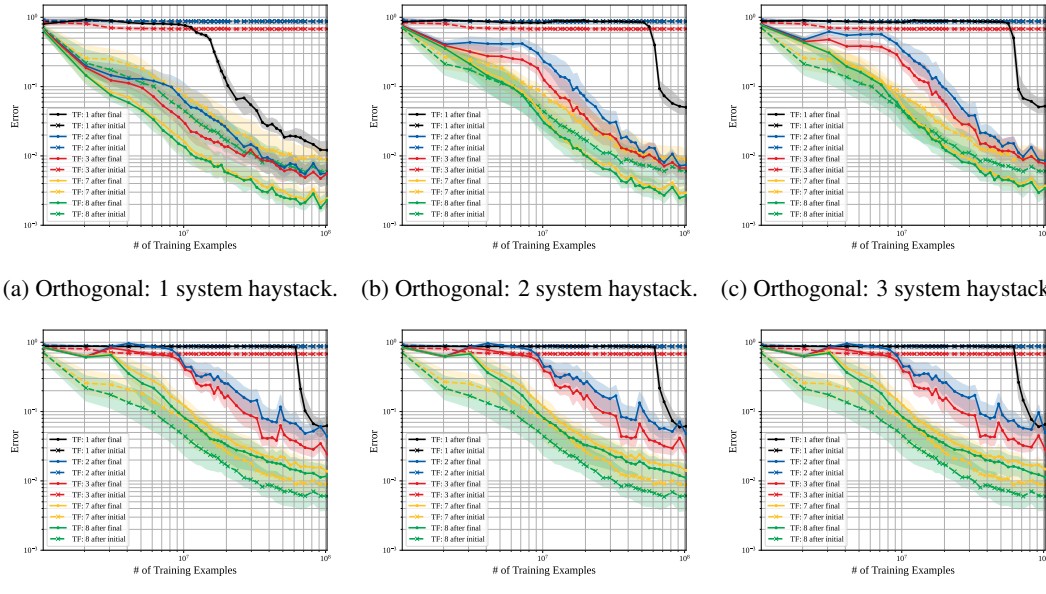

(a) Orthogonal: 1 system haystack. (b) Orthogonal: 2 system haystack. (c) Orthogonal: 3 system haystack.

(d) Orthogonal: 17 system haystack. (e) Orthogonal: 18 system haystack. (f) Orthogonal: 19 system haystack.

Figure 34: Performance of small orthogonal model (701K params) across training — log-scale.

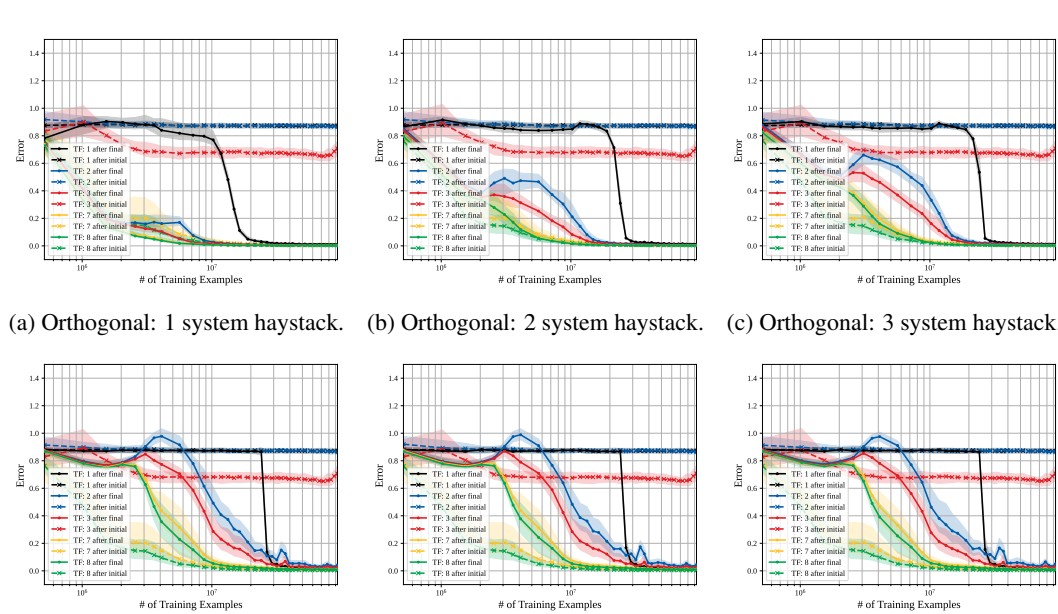

(a) Orthogonal: 1 system haystack.  (b) Orthogonal: 2 system haystack.  (c) Orthogonal: 3 system haystack.

(d) Orthogonal: 17 system haystack.  (e) Orthogonal: 18 system haystack.  (f) Orthogonal: 19 system haystack.

Figure 35: Performance of medium orthogonal model (2.42M params) across training — linear-scale.

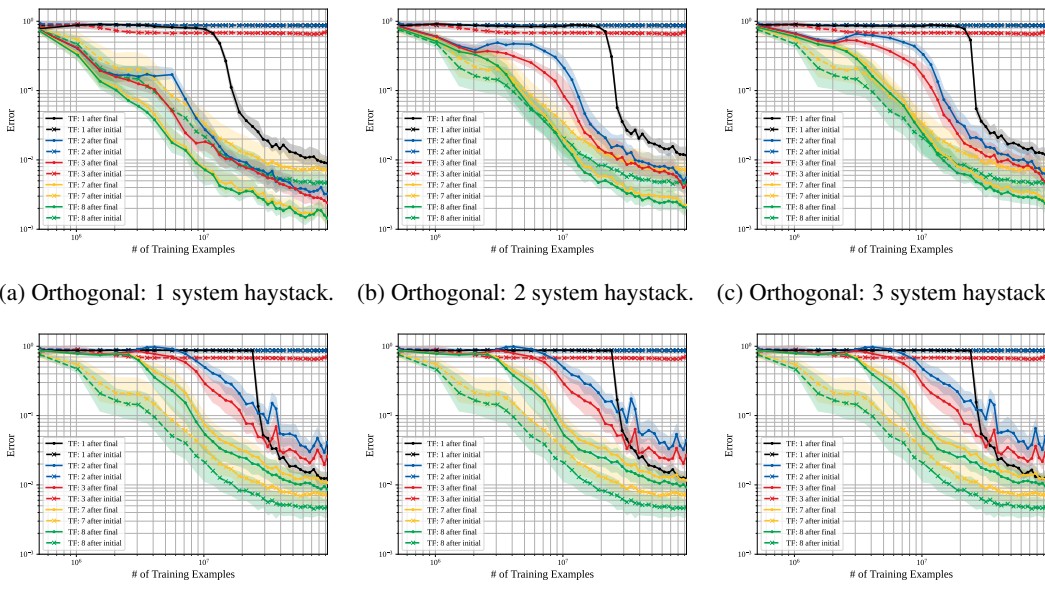

(a) Orthogonal: 1 system haystack.  (b) Orthogonal: 2 system haystack.  (c) Orthogonal: 3 system haystack.

(d) Orthogonal: 17 system haystack.  (e) Orthogonal: 18 system haystack.  (f) Orthogonal: 19 system haystack.

Figure 36: Performance of medium orthogonal model (2.42M params) across training — log-scale.

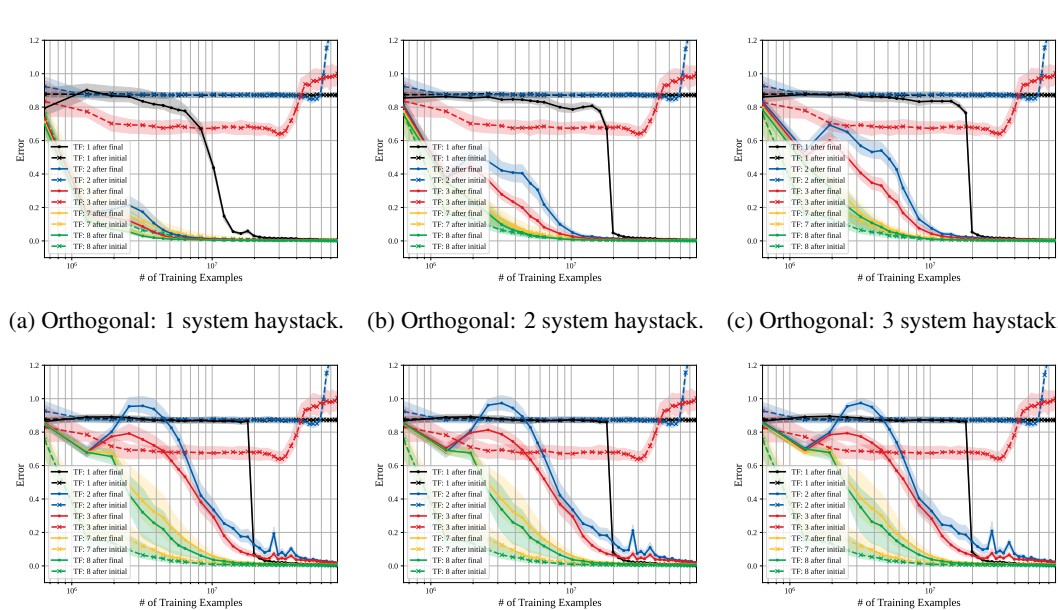

(a) Orthogonal: 1 system haystack. (b) Orthogonal: 2 system haystack. (c) Orthogonal: 3 system haystack.

(d) Orthogonal: 17 system haystack. (e) Orthogonal: 18 system haystack. (f) Orthogonal: 19 system haystack.

Figure 37: Performance of big orthogonal model (10.7M params) across training — linear-scale.

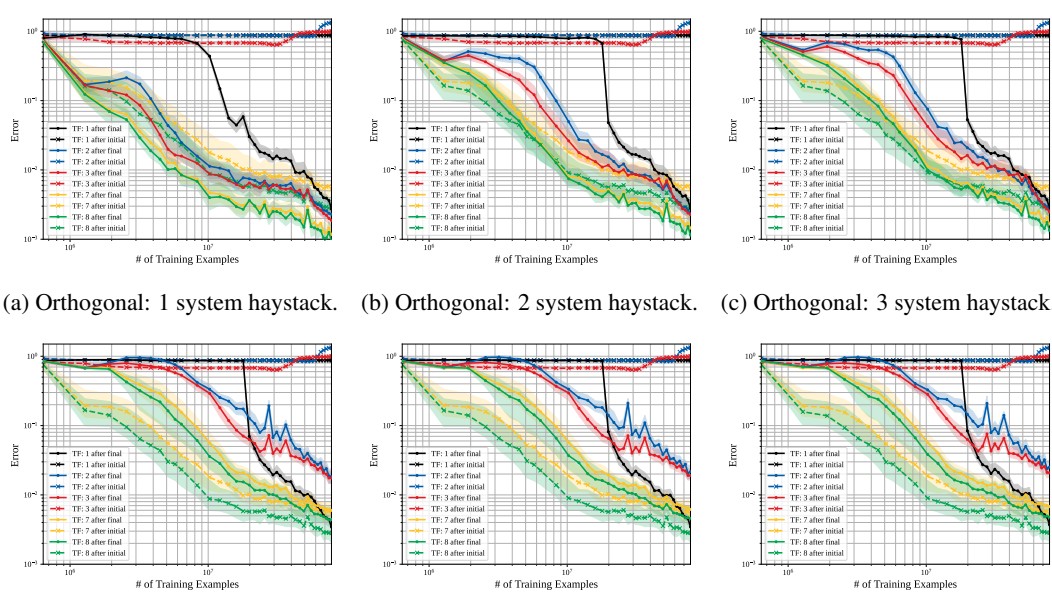

(a) Orthogonal: 1 system haystack. (b) Orthogonal: 2 system haystack. (c) Orthogonal: 3 system haystack.

(d) Orthogonal: 17 system haystack. (e) Orthogonal: 18 system haystack. (f) Orthogonal: 19 system haystack.

Figure 38: Performance of big orthogonal model (10.7M params) across training — log-scale.

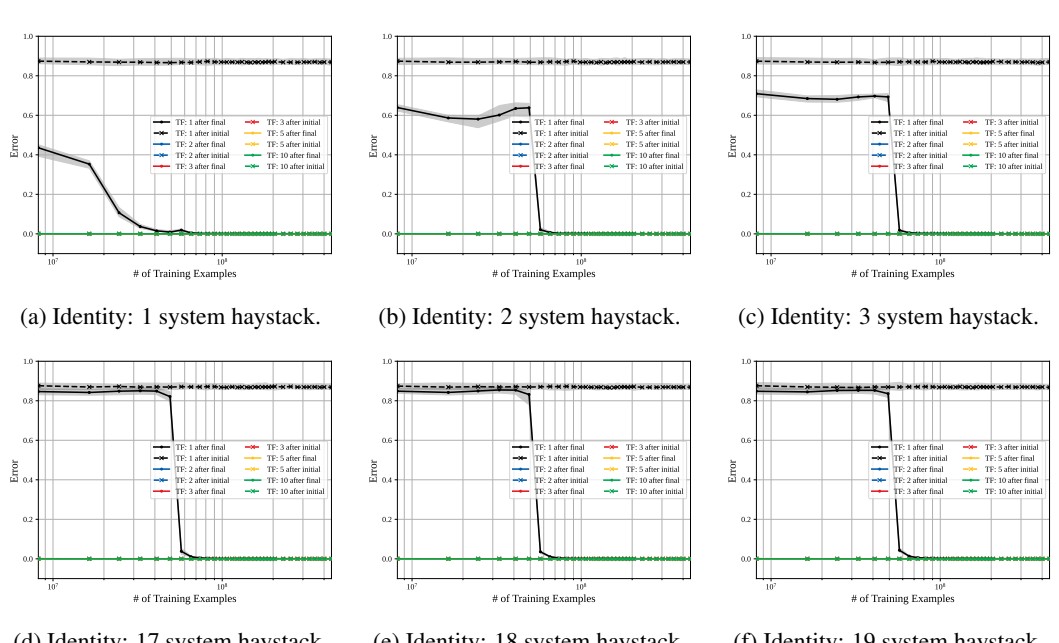

(a) Identity: 1 system haystack.  (b) Identity: 2 system haystack.  (c) Identity: 3 system haystack.

(d) Identity: 17 system haystack.  (e) Identity: 18 system haystack.  (f) Identity: 19 system haystack.

Figure 39: Performance of tiny identity model (212K params) across training — linear-scale.

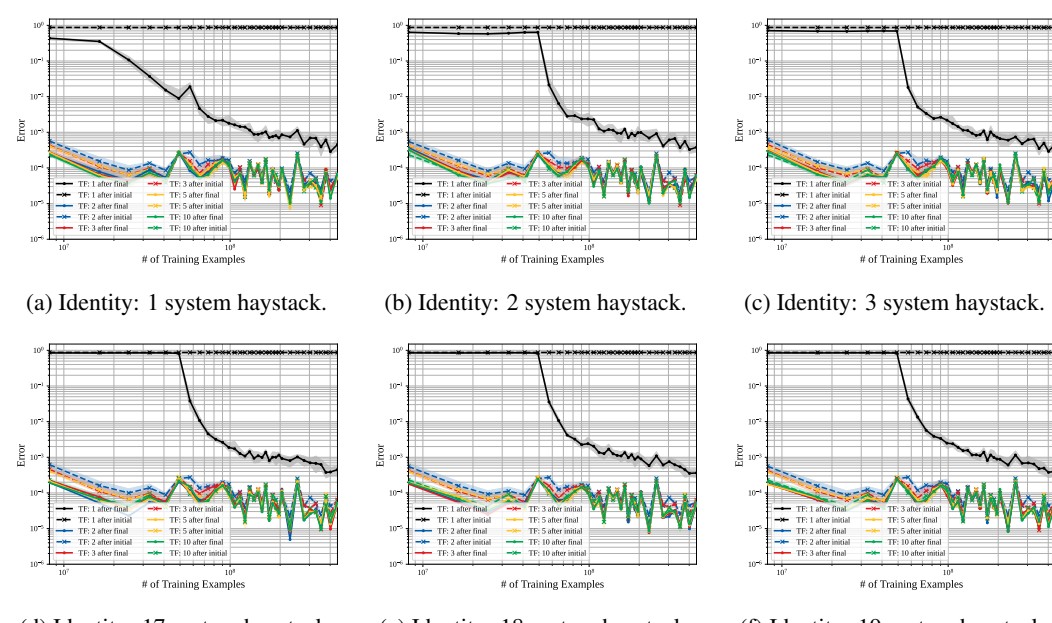

(a) Identity: 1 system haystack.  (b) Identity: 2 system haystack.  (c) Identity: 3 system haystack.

(d) Identity: 17 system haystack.  (e) Identity: 18 system haystack.  (f) Identity: 19 system haystack.

Figure 40: Performance of tiny identity model (212K params) across training — log-scale.

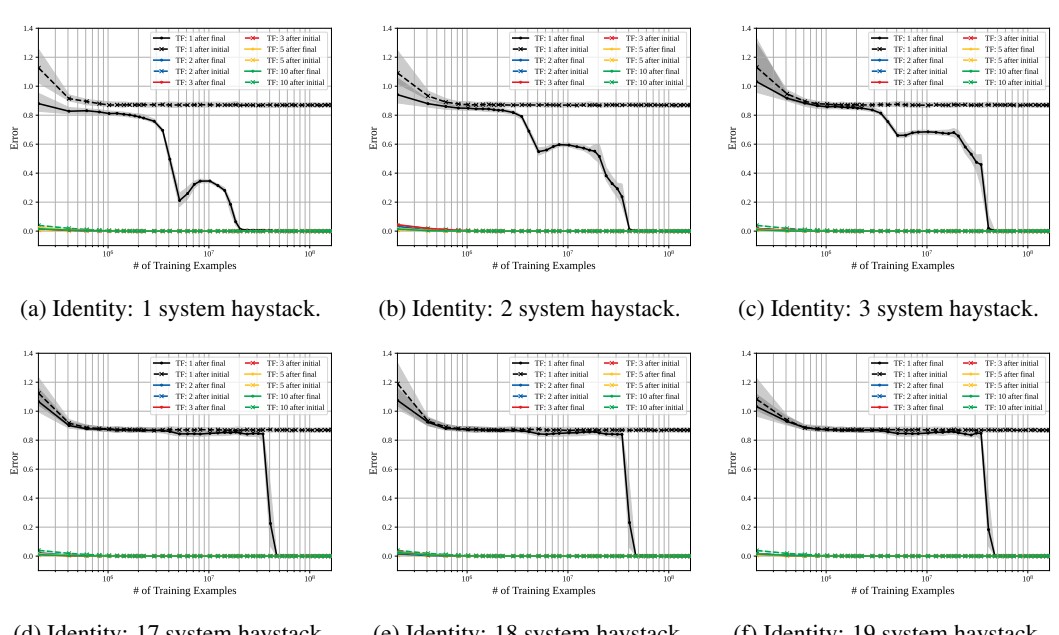

(a) Identity: 1 system haystack.    (b) Identity: 2 system haystack.    (c) Identity: 3 system haystack.

(d) Identity: 17 system haystack.    (e) Identity: 18 system haystack.    (f) Identity: 19 system haystack.

Figure 41: Performance of small identity model (701K params) across training — linear-scale.

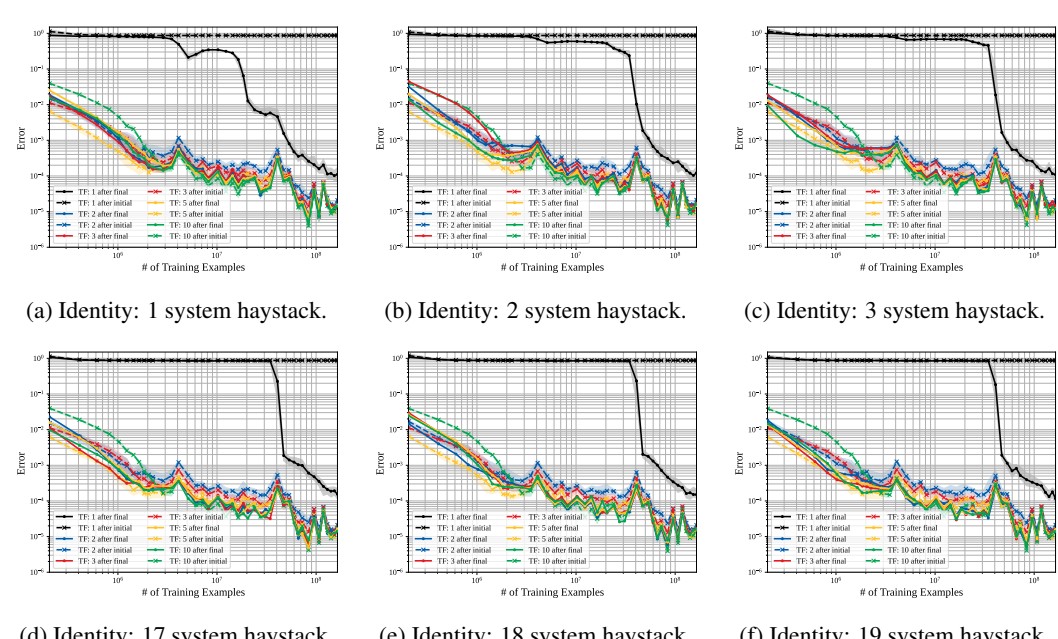

(a) Identity: 1 system haystack.    (b) Identity: 2 system haystack.    (c) Identity: 3 system haystack.

(d) Identity: 17 system haystack.    (e) Identity: 18 system haystack.    (f) Identity: 19 system haystack.

Figure 42: Performance of small identity model (701K params) across training — log-scale.

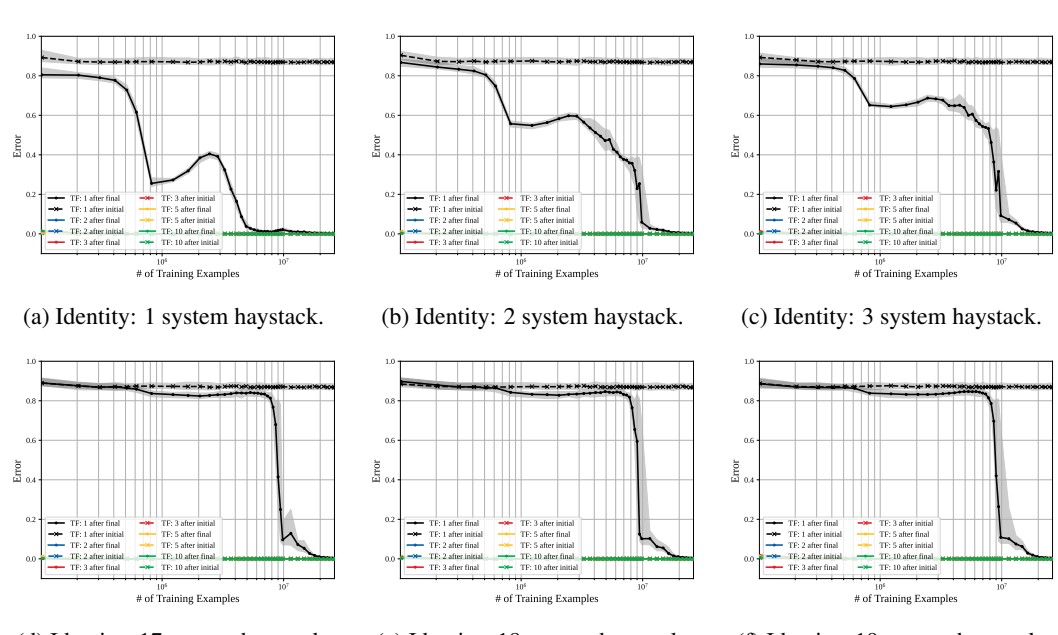

(a) Identity: 1 system haystack.  (b) Identity: 2 system haystack.  (c) Identity: 3 system haystack.

(d) Identity: 17 system haystack.  (e) Identity: 18 system haystack.  (f) Identity: 19 system haystack.

Figure 43: Performance of medium identity model (2.42M params) across training — linear-scale.

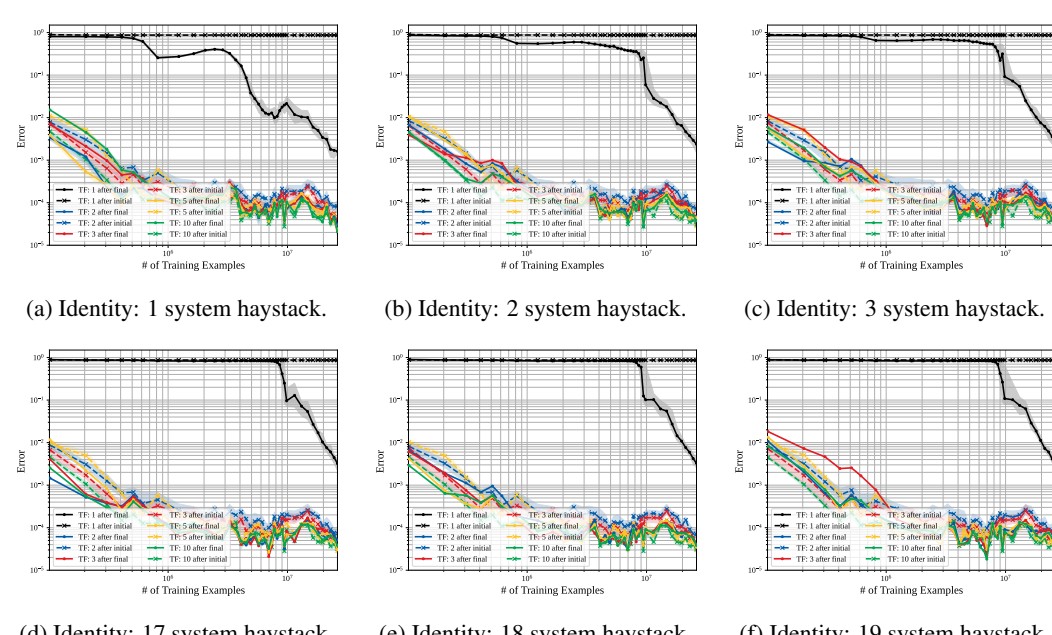

(a) Identity: 1 system haystack.  (b) Identity: 2 system haystack.  (c) Identity: 3 system haystack.

(d) Identity: 17 system haystack.  (e) Identity: 18 system haystack.  (f) Identity: 19 system haystack.

Figure 44: Performance of medium identity model (2.42M params) across training — log-scale.

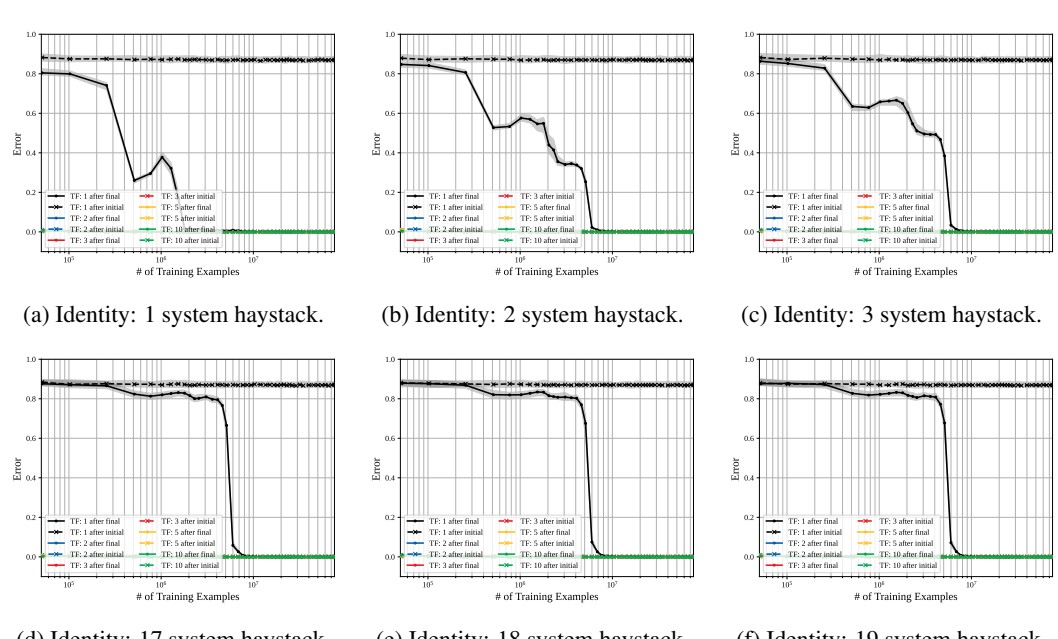

(a) Identity: 1 system haystack. (b) Identity: 2 system haystack. (c) Identity: 3 system haystack.

(d) Identity: 17 system haystack. (e) Identity: 18 system haystack. (f) Identity: 19 system haystack.

Figure 45: Performance of big identity model (10.7M params) across training — linear-scale.

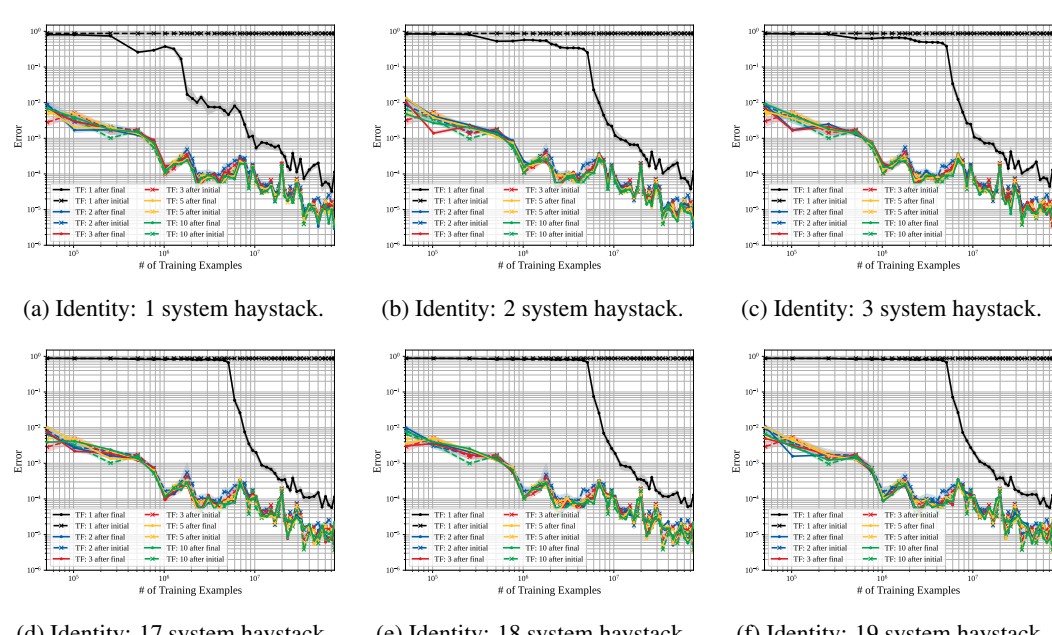

(a) Identity: 1 system haystack. (b) Identity: 2 system haystack. (c) Identity: 3 system haystack.

(d) Identity: 17 system haystack. (e) Identity: 18 system haystack. (f) Identity: 19 system haystack.

Figure 46: Performance of big identity model (10.7M params) across training — log-scale.

## L WHEN DOES THE MODEL LEARN TO RESTART ICL FOR A NEW SYSTEM?

(a) $2.05 \times 10^6$ training examples.

(b) $2.56 \times 10^6$ training examples.

(c) $3.07 \times 10^6$ training examples.

(d) $3.58 \times 10^6$ training examples.

(e) $4.10 \times 10^6$ training examples.

(f) $7.17 \times 10^6$ training examples.

(g) $1.33 \times 10^7$ training examples.

(h) $3.17 \times 10^7$ training examples.

(i) $3.43 \times 10^7$ training examples.

(j) $6.25 \times 10^7$ training examples.

Figure 47: Restarting for a new system — Prediction error vs. the position of a previously unseen system segment in the haystack. Position 1 is the beginning and position 20 is the end. The colored curves correspond to different indices into each system segment.

Continuing on from Section E.2.2, this supplemental appendix takes a more comprehensive look at how the ability emerges for the model to restart its ICL predictions for a new system.

Fig. 47 depicts the prediction error of different model checkpoints on initial segments from previously unseen systems when these segments are at different positions in the haystack. The first segment in

the haystack is denoted position 1 and the last segment is denoted position 20. Furthermore, we look at the prediction error of the model for 1 through 8 indices after the initial open symbol initiating the segment. Subfigure (a) clearly shows that at this early checkpoint $2.05 \times 10^6$, while the model has learned to in-context learn the dynamics for the first segment, there is substantial interference with later segments where performance is poor. Rapidly across the checkpoints illustrated in (b),(c),(d),(e) the model learns how to interpret the symbolic tokens well enough to properly restart its predictions for each new segment and by checkpoint $4.1 \times 10^6$, it is quite good — except at predicting the fundamentally unpredictable second position in a new segment.

Subfigures (f),(g),(h),(i) of Fig. 47 show the more gradual improvements in this ability during much later checkpoints — where the focus is largely on improvements to the performance in predicting the second position in a new segment. Finally, Fig. 47j shows the model's good ability to restart ICL at the end of its training.

## M  FINAL PERFORMANCE WITH POSITION IN SEGMENT AND NEEDLE POSITION WITHIN HAYSTACK

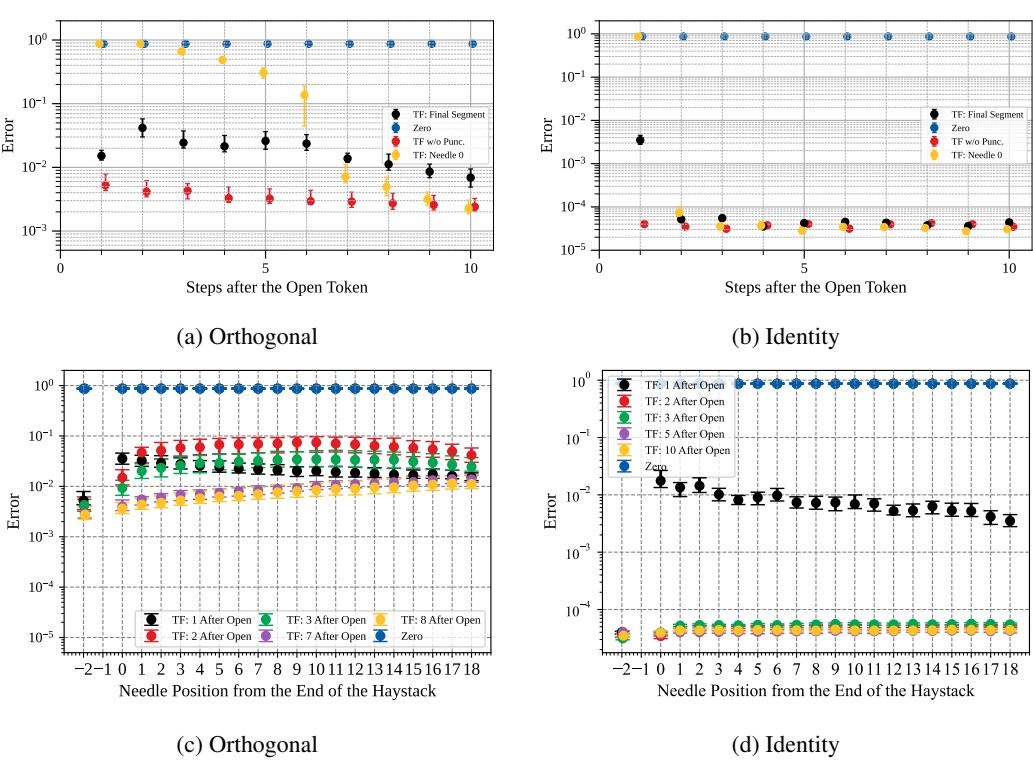

(a) Orthogonal

(b) Identity

(c) Orthogonal

(d) Identity

Figure 48: Comparison of recall performance for orthogonal and identity systems. Figs. 48a and 48b show predictions made in the final test segment (black), from a continuing segment (red), and from the initial segment (yellow). Figs. 48c and 48d show median-squared error vs. the position of the needle in the haystack. For these plots, the x-axis value of $-2$ corresponds to a final segment that is the continuation of the segment before with no interruption by open and close symbols. The x-axis value of 0 is the closest needle position to the final segment while the value 18 is the furthest away. The identity model is after $\approx 1.8 \times 10^7$ training examples and the orthogonal model after $\approx 6.25 \times 10^7$ training examples. "Zero" in the legend corresponds to a predictor that always predicts 0.

Using a haystack of length $N = 19$ for the results in this section, in Fig. 48a, the black dots show the performance of the orthogonal model on the final test segment of a needle-in-haystack sequence. The yellow points represent how well it does on the first system it saw — for which the first point is completely unpredictable and the rest do a bit better. To properly understand recall, we add one more

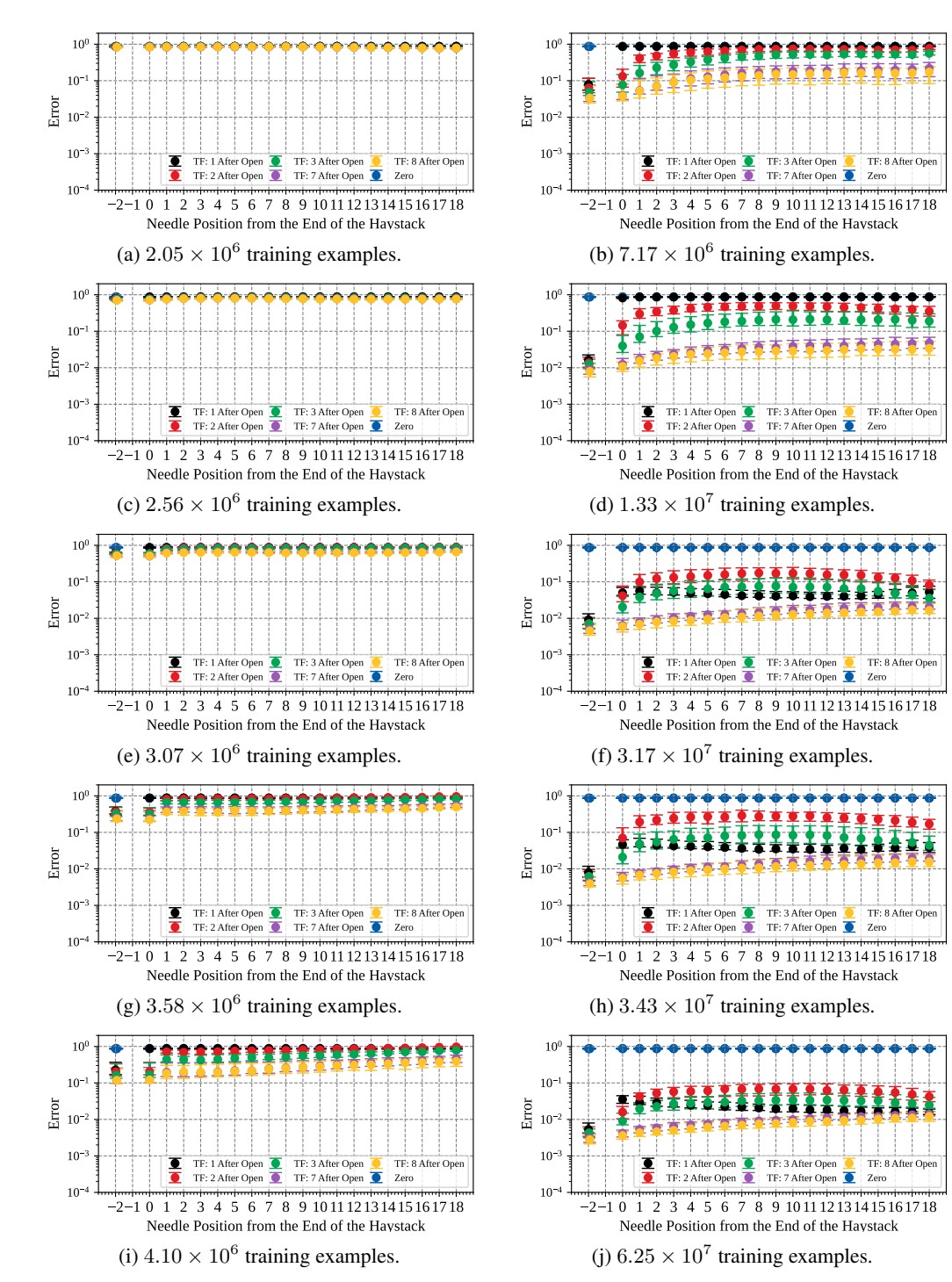

Figure 49: The effect of the needle position throughout training — The median-squared error vs. the position of the needle in the haystack for different training checkpoints of the orthogonal model. For these plots, the x-axis value of $-2$ corresponds to a test segment that is the continuation of the segment before with no interruption by open and close symbols. The x-axis value of 0 is the closest needle position to the test segment while the value 18 is the furthest away. "Zero" in the legend corresponds to a predictor that always predicts 0.

possibility represented by the red dots. This represents the alternative of not terminating the final segment in the haystack at all and simply letting it continue on as the test sequence. We see that the black dots perform worse than the red points for merely continuing the last system in the haystack but better than the yellow points showing what happens the first time we see this system. This clearly shows the ability to do recall, but there is worse prediction performance for the second observation in the test segment than the first (see Fig. 3). The conceptual difference represented by the red dots is that performing well here involves no recall at all, merely being able to continue predicting within one sequence.

Figure 48b shows what this looks like for the identity systems using the black dots. The recall of the constant is a bit spotty in the first position but we get more than 90% of the way there with a squared-error of only 0.01 instead of the 1 we would have if we just guessed the all-zero vector. But after that initial observation in the test segment, the quality of understanding the constant-nature of these segments is quite good with all squared-losses around $10^{-4}$ or below.

We can also see how well the model can use recalled information as a function of the position we are trying to predict within the test segment. This parallels the natural language questions in (Liu et al., 2024b) that spawned widespread "needle in haystack" testing for LLMs generally (Kamradt, 2023). Figs. 48c and 48d show the effect of the needle position on the prediction performance of the orthogonal and identity models respectively. Here, we can see the first observation prediction quality in black showing higher quality when the needle is in the earlier positions within the context (larger position values in the figure). This is somewhat different from the "U-shaped" curve found for language-based LLMs although both have better performance when asking about the first thing seen. Meanwhile, the observation prediction quality for the second position is actually slightly better if the needle is closest to the test segment.

In Fig. 49, we look at how the effect of the needle position develops throughout the training of the orthogonal model. Again, the median squared error of its predictions on different indices in the test segment are shown for 19 needle positions for 10 different training checkpoints. For indices 7 and 8 in the test segment, the trend of better predictions for closer positions to the test segment is the first trend to develop in training and is sustained throughout. This same trend is also seen in indices two and three after $4.10 \times 10^6$ training examples have been processed, but by $3.17 \times 10^7$ training examples the model's predictions for the needle positions furthest away from the test segment start to improve for indices one, two, and three.

