# OpenReview forum: "Decomposing Prediction Mechanisms for In-context Recall"
_ICLR.cc/2026/Conference — Submitted to ICLR 2026_

### Official Review · Reviewer_Rust · 2025-10-24

**Soundness:** 3
**Presentation:** 1
**Contribution:** 2
**Rating:** 6
**Confidence:** 3

**Summary:**

In this paper, the authors aimed at creating a family of toy models for exploring the known challenge of long-context learning for LLM. The proposed toy model have different time series data interleaved with distinct labels. The authors found that LLM developed two distinct learning mechanisms in performing next token prediction on the toy model. The first mechanism focuses on identity regime change in the data, and the second one perform next token prediction based the data observed. The two mechanism also seem to follow different learning dynamic, and the second one developed earlier than the first.

**Strengths:**

1. The author aimed at a crucial problem in understanding LLM, namely the challenge of  challenge of long-context learning.
2. The designed toy model indeed is simple in structure on one hand, but capturing some nature of human languages on the other hand.
2. Quite extensive numerical experiments are conducted.

**Weaknesses:**

1. The main message the author intended to convey is not very clearly presented. It appears to be the discovery of the capability of Transformers on developing distinct mechanisms for predicting different token positions in a single task via a study of a specially designed toy model. Although related statements in various places of the paper do not seem to always be precisely the same.  While the first two hypotheses are shown not to hold, the language in the description of the conjecture and its confirmation is very vague and puzzling.  Moreover, to confirm or deny such s strong conjecture, a much more thorough set of experiments needs to be designed and carried out, not simply an observation continuation and new initiation can not be distinguished.

2. The connections between the observations, existence of distinct learning mechanism to the challenge of long-context learning are not explicitly stated.

**Questions:**

1. I feel that the critical question is that why and how the understanding of this toy model can help us understand the capability of long  long-context learning.of LLM. I don't see this is addressed in the paper.
2. It appears that at any fixed time of learning, there will be much more time series data than discrete symbolic labels, is that the reason that the second mechanism develops much earlier than the first.
2. How are the issues presented in Sec. 5(discussion) related to the problems and contributions presented in introduction? I failed to see the clear connections.

---

> ### Author Response · Authors · 2025-11-25
> **Why is this interesting for long-context?**
>
> Thank you for your review and your questions.
>
> Let’s start with the reviewer’s main question: what does this paper have to do with long context learning in LLMs? The key observation is in Figure 3 of the paper as we vary the number of distinct systems present in an ever longer context. At first glance, one notices curves that get worse as the context gets longer. By itself, that is not surprising — this is why we care about getting long-contexts to work in LLMs, because it doesn’t just happen. It is with a closer look that we see the surprising pattern: the black dots corresponding to the 1-after-query task are actually quite stable as we increase the number of systems in the haystack (along with the context required to hold that much), it is the other curves that are getting worse.
>
> This is seemingly paradoxical: intuitively, the 2-after-query task is easier than the 1-after-query task since the 1-after-query task requires the model to recall both the previously seen state of the system along with the previously ICL-learned system dynamics and then apply those dynamics to that recalled state. Meanwhile, the 2-after-query task merely requires applying those recalled system dynamics to the currently-visible state in this token itself. And yet, the quality of performance is reversed when contexts get long: 2-after-query performs notably worse than 1-after-query. This cannot be explained by a wildly different proportion of seeing these two kinds of tasks in the pretraining data — essentially every time we saw a 1-after-query recall in the pretraining data we also saw a 2-after-query recall.
>
> The paper provides one part of the explanation — the model has actually learned two different mechanisms for doing these two intuitively tightly-related tasks. The out-of-distribution experiments in the paper establish this. And somehow, the mechanism learned for the 2-after-query recall (along with everything afterwards), is far more sensitive to having more context than the mechanism for doing the 1-after-query recall.
>
> Is this relevant for long-context learning? Yes! Because it provides a toy-world replication of a phenomenon well-appreciated by practitioners: that just because a model can do clean associative recall like what MQAR and traditional needle-in-a-haystack tests benchmark, it doesn’t mean that it can actually use long contexts well in practice. There’s a gap. Our paper essentially conjectures that this is probably due to whatever inductive bias causes observation-based recall to emerge in our toy world.
>
> In response to the reviews, we ran an additional out-of-distribution experiment whose results are in the new Figures 6 and 7 in the Appendix. Here, we made every trace in the haystack identical to eliminate informational interference between traces. When we look at Figure 6.e (with 19 different traces in the haystack) as compared to Figure 6.f (with 19 identical traces in the haystack) we see a substantial difference for the 2-after-query and 3-after-query performances: they are both better and flat across which position is being queried. Meanwhile, the 1-after-query performance is essentially identical in shape and does not improve by having every trace be identical. This not only strengthens the argument that there are distinct mechanisms, but it also tells us that these different mechanisms experience informational-interference very differently. If the answer was something simple like repetition strengthens recall, then it would also help the 1-after-query performance — but that one isn’t helped.
>
> This is not to say that there isn’t an effect of where information is located in the context — Figure 7 makes that amply clear as we plot different dots corresponding to different positions in the haystack while using the x axis to increase the number of traces in the haystack. However,  Figure 7.a clearly shows that informational interference vis-a-vis the k-after-query mechanism (for k>1) is steadily degrading performance while this is not as much the case for the 1-after query mechanism.
>
> We will make the above case clearer in the final version of the paper.

---

> > ### Author Response · Authors · 2025-11-25
> >
> > In terms of the other questions, we don’t yet know the answer as to why the observation-based mechanism develops earlier than the label-based mechanism. Rather than the number of symbolic labels per-se in the data, we think that one natural hypothesis that must be considered is that it has to do with how often a 1-after-query type recall event occurs in the training data as compared to k-after-query for k>1 but k<7 (at which point relearning the system would be as good as recalling it) — that would be a factor of about 6. This doesn’t exactly match the difference between when the mechanisms emerge and it also doesn’t explain the very different shapes of emergence during training. This is an important problem and we hope that our toy world helps the community make progress on it.
> >
> > Finally, we wrote the short Section 5 (discussion) not as a recapitulation of contributions or as a continuation of the introduction. Instead, we just made additional comments connecting to training dynamics speculating on how multiple distinct mechanisms for closely related problems might persist. If indeed the persistence of such more sensitive and less robust to informational interference mechanisms is one of the reasons why long context learning in practice is worse than what simple needle-in-the-haystack recall tasks might suggest, we wanted to encourage community thinking about how one might improve training recipes to help improve things.

---

### Official Review · Reviewer_7gvh · 2025-10-31

**Soundness:** 4
**Presentation:** 4
**Contribution:** 3
**Rating:** 6
**Confidence:** 4

**Summary:**

ICL is a well studied phenomenon in the ML community. Various tasks, such as MQAR and regression, have been proposed to test the ICL capabilities of models in the past. The beauty of each is it both tests the model's ability to perform lookup operations (MQAR) and more complex operations only depending on the previous token (regression). This work combines these into a task using linear dynamical systems, where each system is marked in-context by a specific query label. Two observations are seen: the model uses the open-query label to perform the correct task, and the model uses past elements in the sequence to continue the task. These observations are validated by configuring the systems and states to align, allowing for a clear test of these observations in a controlled setting. Further investigating that these different mechanisms exist within these learned models, a mechanistic study is conducted separating out two circuits from within the model that have markedly distinct performance on the two different subtasks of recall and execution.

**Strengths:**

- This new tasks used to test ICL is appealing. It provides a nice link between standard ICL problems while keeping almost everything continuous, hence interpretable. The dynamics are quite intuitive yet retain significant depth to make an interesting analysis.
- The experiments investigate a variety of different interventions to test the hypotheses H1 and H2 and show, clearly, that models will learn to perform the correct task on the first token after the new-task identifier, and then relying on previous outputs to generate more tokens.
- The results regarding the disparity between 1-after and 2-after display a very interesting aspect of how these models learned to solve a task composed of a mixture of regression and associative recall
- The circuit analysis added depth into the difference between these two mechanisms as two truly separate aspects of the model.
- The writing is very clear, with claims and hypotheses which are most relevant to the reader highlighted in boxes, along with the distinct mechanisms all given separate colors for the reader to decern them.
- The toy model architectures are cleanly described in Appendix I.

**Weaknesses:**

- Much of the work (specifically the figures) is focused on the training dynamics of these model. While interesting, and should certainly be highlighted, the claims of the resulting model having these two distinct mechanisms to understand these linear-dynamics inputs typically are most important at the very end of training. This wasn't particularly central in the played results and rather had to be pulled out from the training dynamics
- The paper focuses on one specific task and found a property about ICL performance on this specific hybrid task. There was not any investigation into whether these same behaviors can be seen with other tasks (possibly other hybrid tasks), resulting in a possibly narrow applicability
- A few (and only a few) task design choices were not clearly described (see questions)

**Questions:**

- Is there any reason for selecting 5 dimensions? Did this coincide with the model able to learn it at the scales tested, where too much greater led to bad performance while any smaller made the task incredibly easy?
- Why train OLMo specifically and not some other language model like LLaMa? Was there any investigation into these tasks trained using different foundation models?

- Why specifically this setup? Did the close tokens help the model find the last token in the output of this current sequence? Why not always use the last token of any of the systems to be the input of the next one?

---

> ### Author Response · Authors · 2025-11-25
> **System dimensions, open-and-close tokens, and why independent traces**
>
> Thank you for your thoughtful review and insightful questions.
>
> In terms of your specific questions, we selected 5 dimensions merely as a default that we never changed. The Du, et. al. paper on learned Kalman Filtering, whose codebase we built upon, used five-dimensional observations. Initially, we wanted to make sure that our results were easily comparable to theirs and so wanted to keep as many things the same as possible. Eventually, while the technical focus of our work shifted, this initial choice of 5 was never revisited because it never seemed to be either limiting or critical to the story. We can run fresh training runs (with our limited compute access) as we vary the system dimensions and can include such results in the appendices of the final paper. But we couldn’t run these by the time we’re responding to the review. Our suspicion is that nothing much would change if we went smaller until we hit 2 dimensions at which point the manifold of orthogonal matrices is just one-dimensional (rotations and reflections) — when it is that small, random samples can land close to each other when we start drawing many systems at random. Larger systems will presumably at some point start taxing the expressivity of our embedding space and the attention heads, and potentially also the limited number of layers in our model. It would be interesting to map this out, and we hope that our paper will encourage this kind of mapping since it would add to our appreciation of the fundamental role of depth and width in LLMs.
>
> In terms of our setup, we chose to use both open and close tokens to punctuate the context because (a) Our mental model was different kinds of parentheses; (b) We’d seen lots of code-oriented formats having matched delimiters; (c) As the reviewer suspects, we guessed that it would make it easier for the model to find the relevant last token from the previous occurrence of this sequence and that the model might actually store the relevant state in the key,value pairs corresponding to the end parenthesis due to causal masking. Of course, when we made this choice, we were not expecting there to be any fundamental difference between the 1-after-query and 2-after query mechanisms. That was a surprise to us — since our intuition (c) above was based on the idea of using the same way of recalling the relevant information.  After seeing your review, we think that it would be informative to run an out-of-distribution experiment where we use mismatched open and close tokens in the haystack to see what happens when we query in all possible ways: actual open, open-corresponding-to-actual-close, actual close, close-corresponding-to-actual-open, as well as misdirection versions of the same. We hope to include that in the final version of the paper to shed even more light on what is going on here and doesn’t involve doing another training run. It would also be interesting to try a fresh training run with only open tokens as well as one with only close tokens to see if those are any different in their behavior.
>
> The other question you raise is easier to answer. The reason we didn’t chain the states together across different systems (i.e. feed the last token/state from one system as the input to the next system) is because we wanted to model/caricature the kind of associative recall that one might have in a RAG system where there are many different potentially relevant snippets in the context, but most of them are completely irrelevant to what we want to do. If we had chained states together, then the relevant last-state would come from the immediate past while only the relevant system dynamics would have to be recalled. (Another reason is that the approach we took allowed us to do training runs with an even simpler model — where the system is always the identity map and the different segments are just constant vectors. Those training runs showed the same emergence of associative recall and internally, we felt it was important to have a problem that looked as close to classic MQAR as possible. We didn’t report on the constant case in the submitted paper because it wasn’t relevant to the story.)  That said, it would be nice to see what happens in this case where only the system dynamics are being recalled instead of both the dynamics and the state.

---

> > ### Author Response · Authors · 2025-11-25
> > **Why OLMo and plot focus on training dynamics**
> >
> > We fine-tuned OLMo training checkpoints instead of LLaMa simply because the OLMo model had publicly available pretraining checkpoints and the 8B model was clearly capable enough at the end of training to perform our ICL translation task. We only tried OLMo2 and not any of the other larger LLMs with open training checkpoints like Pythia, Amber, Crystal, K2-65B, Moxin, etc. or those that have emerged more recently. We agree that it would be interesting to see the pattern we conjectured (i.e. separate and earlier emergence of the “continue” ability as distinct from the “initiate” ability for an ICL-specified task) happen in multiple open-weights open-training-checkpoints.
> >
> > This last point brings us to the reviewer’s comment on our focus on the training dynamics. It’s true that all the figures we have are focused on the training dynamics — this was because progress in training was a natural horizontal axis. The very different behavior during training was also what allowed us to suspect that there were different mechanisms in play at all. Having seen the different behavior during training, we were able to use out-of-distribution experiments to verify that indeed the mechanisms are different. Crafting and interpreting these out-of-distribution experiments required us to use the specifics of our toy world’s tasks. However, for verifying that the same phenomenon exists in LLMs, it was easiest to first look at training checkpoints and the English-to-Spanish translation task was the first and only thing we tried. We hope that our paper prompts others with more NLP experience to push on this direction and explore both other tasks as well as variations that can function like the out-of-distribution experiments we chose in our toy world’s tasks. We agree with the reviewer that it would be interesting to understand this.
> >
> > Finally, the other reason we use training dynamics is because the story can be a bit subtle. We don’t emphasize this in our writing, but Figure 22 in the Appendix of the paper shows an odd behavior for the out-of-distribution experiment of misdirecting to an unseen sequence — after first behaving for a long while like misdirecting to an incorrect seen sequence, the behavior of the 2-after-query and 3-after-query predictions “phase-transitions” to something different that is clearly very different. If we had only looked at out-of-distribution experiments at the end of training, this wouldn’t have been visible.

---

### Official Review · Reviewer_sGTe · 2025-10-31

**Soundness:** 2
**Presentation:** 2
**Contribution:** 2
**Rating:** 2
**Confidence:** 3

**Summary:**

This paper studies mechanisms through which transformers can perform in-context prediction.
In models trained on a novel synthetic task, the paper discovers two mechanisms ("label-based" and “observation-based”).
A further experiment on OLMo checkpoints provides further evidence from a translation task.

**Strengths:**

- The paper provides a new family of well-specified toy problems to study mechanisms used in Transformers for in-context recall

**Weaknesses:**

- The setup, as motivated in Section 1.1, appears quite specific. I was missing a motivation of why the setup is of broader relevance or interest, e.g. to language models, or the transformer architecture, etc. This is a concern especially as the paper mainly concerns empirical studies of toy models trained on a toy task.
- Interpretation of Section 3: Section 3.3, line 400: "0% edge overlap between the 1-after query and 2-after query circuits": As far as I understood the description in the section, the circuit finding strategy used here imposes no pressure towards overlapping circuits. Hence, it is conceivable that the reason for 0% edge overlap is just that the model has multiple redundant mechanisms for the two tasks, and the circuit finding algorithm happened to find different mechanisms when run on the two tasks. I'd appreciate if the authors can comment on this.
- Section 4: I didn't understand the task used here. On the one hand, the task is English-to-Spanish translation, on the other hand "we also change our analogous natural language setup to have in-context labels with no semantic meaning" What does this mean? How does this relate to the examples given in Appendix G?

**Questions:**

I'd appreciate if the authors can clarify if any of the weaknesses listed above may result from misunderstandings on my end. I'm happy to reevaluate my score on the basis of the response.

---

> ### Author Response · Authors · 2025-11-25
> **Relevance and the task used with OLMo2 checkpoints**
>
> Thank you for your review and your questions. To address your concern regarding the relevance of our setup, our goal was to construct a minimalist toy world that captures three key features of large language models: in-context learning (ICL), true sequentiality (i.e. tokens depend on nontrivially on prior ones), and associative recall (i.e. the relevant section of past context can change — it isn’t always the immediate past). Prior work had shown vector least-squared problems can be used to elicit/study ICL, although without either sequentiality or associative recall. Using vector-valued time-series lets us have true sequentiality and interleaving different time-series punctuated with symbolic labels lets us have true associative recall. The value of the toy world is to be able to train small models (millions of parameters) using perfectly understood (and controllable) synthetic data, as well as being able to compare to known analytic baselines. The toy world also creates more opportunities for careful out-of-distribution experimentation.
>
> Why study a toy at all? Because presumably, transformer-based neural-network models have their own implicit biases in terms of what kinds of patterns they learn and how they learn them. These implicit biases of the neural-net architecture (along with the optimizer, training recipe, etc.) are not about language. Studying what transformers do in a toy world helps us discover new insights by suppressing the full complexity of actual natural language and permitting full training experiments that would be out-of-reach for actual language given the compute scales required.
>
> Indeed in our paper, we found a surprising pattern in our toy world — namely that a different mechanism seems to be involved in initiating an in-context-learned task as compared to continuing it. This pattern was first found by examining what happens during training — the “continue” ability emerges much earlier in training. However, we were also able to perform definitive out-of-distribution experiments to confirm that the “continue” ability seems to ignore the content of the symbolic labels used to initiate the task.
>
> To verify that the new pattern we discovered represented something about transformers and was not just an artifact of our idealized toy world of interleaved time-series, we used OLMo2 training checkpoints along with an English-to-Spanish translation task in Section 4 and Appendix H. This was a few-shot example (two example sentences in English and Spanish prefaced by “X:” and “Y:” each as shown in the Appendix) given in the prompt followed by a test sentence to translate. And indeed, we saw the ability to continue a Spanish translation emerged earlier than being able to in-context implicitly understand that the symbolic label “X:” meant “English:” and that the symbolic label “Y:” meant “Spanish.” The use of meaning-free symbolic labels like “X:” and “Y:” was important here because this requires the meaning of the labels to be understood using ICL. By also running an experiment using meaningful labels like “English:” and “Spanish:” (as described in the Appendix), we could see a difference — meaningful labels allowed the task initiation to happen earlier in training (presumably before the ICL ability had emerged).

---

> > ### Author Response · Authors · 2025-11-25
> > **Pruning**
> >
> > As far as the edge-overlap findings go, we originally had run only one pruning experiment with only one random seed using essentially the default settings and had found 0% edge overlap between the circuit that did the 1-after-query task and that which did the 2-after-query task. We had imposed no pressure towards either overlapping circuits or nonoverlapping ones. We had originally done this edge pruning to see whether there was any mechanistic plausibility to what we were seeing in our out-of-distribution experiments — the seeming presence of entirely distinct mechanisms for performing tasks that we had considered quite related to each other. We acknowledge that circuit discovery algorithms are notoriously difficult to purely isolate as faithful to the circuit used for the task (see Miller et al. https://arxiv.org/abs/2407.08734) and will be including such caveats in the final version.
> >
> > However, in response to the reviews, we reran the circuit pruning as we swept through different values for the optimization knob that controls sparsity to range sparsity from 1000 found edges down to about 10 found edges. This is visualized in Fig 8 in the revised pdf. Then, we looked at the edge overlaps within those found for the 1-after-query task, within the 2-after-query task, and then across the 1-after-query task and the 2-after-query task. The level of overlap within 1-after-query circuits (found at different levels of sparsity using fresh runs with different randomness) was very high as you can see in Fig 8. Similarly, the level of overlap within 2-after-query circuits was also very high. However, the level of overlap across any 1-after-query circuit and any 2-after-query circuit was very low (although not always exactly zero). So while we cannot definitively rule out very redundant but highly overlapping circuits between the 1-after and 2-after tasks, the huge discrepancy between the internal overlap within multiple random runs for the same task and across multiple runs for the different tasks makes that very unlikely since such a situation would manifest as low overlap for different sparse circuits for a single task as well.  Within our paper’s story, the edge-pruning results are just saying that indeed it is possible to approximately do these tasks using non-overlapping circuits in the learned network. It is the out-of-distribution experiments that are actually convincingly showing that the mechanisms are functionally different from each other because they behave differently.

---

> > > ### Comment · Reviewer_sGTe · 2025-11-28
> > >
> > > Thanks a lot for your response. I'll respond in more detail as soon as I get a chance.
> > >
> > > The EDIT button is currently deactivated, but I'll update my score to 4 when it becomes reactivated.

---

### Official Review · Reviewer_hXkf · 2025-11-01

**Soundness:** 3
**Presentation:** 2
**Contribution:** 2
**Rating:** 4
**Confidence:** 4

**Summary:**

This paper proposes a new methodology to study in-context behaviors in transformer models. They create a sequence which consists of segments of observations drawn from different distributions. Each segment begins with a special token, termed "symbolic punctuation label" (SPL), so model must choose between inferring the next observation based on the SPL or the observations in the context. They provide experimental evidence suggesting that the latter choice develops earlier in training than the first.

**Strengths:**

The paper proposes a new synthetic needle in a haystack task, which is interesting and novel. The experimental design of using 1) misdirected SPL and 2) synchronized observation are well-motivated.
They also extend their analysis on OLMo checkpoints on translation tasks, which ground their findings with real-world evidence.

**Weaknesses:**

I find that many of the claims in the paper could be considerably strengthened and simplified.
* I am not fully convinced that the label-based recall hypothesis (H1) is decisively ruled out. One could explain Figure 1b) simply from the fact that the model sees more observation tokens (the 1-after and 2-after query) than the open SPL token itself. It would be valuable to test whether increasing the representational weight or length of the SPL (e.g., by replacing each open-label with a multi-token sequence or embedding-enlarged symbol) causes the model to rely more on label information.
* Figure 1a is not clear. the query and misdirection tokens (the parentheses) are identical.
* The results on observation-based recall degrades with more systems (relative to the performance of 1-after query) is quite surprising and not well explained. Section 3 devotes much time to validating H1 and H2, but in my opinion does not spend enough discussion on how to explain the phenomena in Figure 3. For example, whether it reflects interference among observation traces, reduced signal-to-noise in embedding space, or limitations of attention span. A deeper discussion or ablation (e.g., varying the degree of interleaving or the correlation among systems) would strengthen the empirical interpretation.
* The paper concludes that “the model mostly leverages mechanistically different learned mechanisms for consecutive tokens,” but it remains unclear how many mechanisms exist in total (two, or more?). The pruning analysis isolates only two (corresponding to the 1-after and 2-after query positions), and while these show 0% edge overlap, this alone does not establish that the model’s behavior decomposes neatly into exactly two circuits. Clarifying the scope of this claim would strengthen the mechanistic argument of the paper.

**Questions:**

The paper would benefit from an ablation on the length of each interleaved segment. Since segment length determines how many observation tokens are available for inferring each system’s dynamics, varying it could reveal whether the distinction between label-based and observation-based recall arises from token exposure rather than a fundamentally different mechanism. For instance, longer continuous segments might strengthen observation-based continuation, while shorter segments could force greater reliance on symbolic labels.

---

> ### Author Response · Authors · 2025-11-25
> **Segment-length (non)effects and Interference as the source of degradation for observation-based recall**
>
> Thank you for your careful reading and insightful comments. Regarding Figure 1a, that was a typo in our diagram. Below the misdirection arrow there should have been a purple open curly bracket. Thank you for catching this error. We have corrected it in Fig.1a in the uploaded revision PDF.
>
> **Segment length ablations**
>
> To answer your most immediate question, we include some experiments with shorter segments in the haystack (trace lengths 5 and 2, along with 15 instead of just 10 in the original submission). Trace length 2 is the minimum since one needs to see at least 2 observations in a trace to be able to make any prediction with mean-squared-error better than just predicting a zero vector. The new figures are in Figure 5 in the updated pdf.
>
> Even in these cases, the misdirection experiment unambiguously shows that while the “1-after” predictions (i.e. those for the value immediately following a query) do use the content of the symbolic labels, those for 2-after and 3-after the query symbol do not use the content of the symbolic label to determine what system to continue — the performance is unaffected by the misdirection. For example, Fig 5.a (no misdirection) vs Fig 5.b (misdirection) shows what happens with traces of length 2 in the haystack. The black 1-after curve responds to the misdirection and moves up in Fig 5.b once recall emerges instead of moving down in Fig 5.a. Meanwhile, the blue curves for 2-after performance and red curves for 3-after are unchanged.
>
> This behavior is the same as what we observed in the paper with 10-long segments (replicated here in Figs 5.e and 5.f for convenience) — and strengthens our claim that different mechanisms are operational for the different tasks of initiating a recall and continuing one. Although the reviewer’s suggestion is plausible that the extent of value-token exposure as compared to symbolic-token exposure in context could modulate what would happen, that doesn’t appear to be the case for our experiment. We agree with the reviewer that understanding exactly *why* this is the case is interesting, and we hope that our paper will help seed such investigations.
>
> **Observation-based recall degrades with more systems in the haystack**
>
> We have run further tests to investigate two hypotheses for the degradation of observation-based recall with more systems in that haystack. The first hypothesis is that degradation is due to interference between observation traces (Here, we’re folding together the reviewer’s interference and embedding signal-to-noise suggestions because signal-to-noise reduction is a way that interference can operationally manifest.). The second hypothesis is that the degradation is because the model needs to recall a system that is further away in the context (this is what the reviewer suggested as limitations of attention span) from the test segment as more systems get added to the haystack.
>
> To test the interference hypothesis, we ran another out-of-distribution experiment where every haystack segment was an identical trace, yet they all were labelled with different SPLs as if they were all coming from different systems. The test segment is then a continuation of this one identical trace. Thus there is no information interference between segments, no matter how many there are. In this new experiment, the performance (shown in Fig. 7.b for identical-trace haystacks) on the 2-after query (blue) and 3-after query (red) predictions no longer degrades substantially as more haystack segments are added as it does for non-identical traces in Fig 7.a. Furthermore, the prediction performance is better than what it is when there are distinct systems within the haystack and is always better than the pure recall performance of 1-after query. (Note, in this experiment every one of the traces in the haystack is identical and so one might expect the performance of the 1-after query to also improve — but this is not the case. Instead, the performance closely tracks what happens where there are distinct systems in the haystack.) Therefore, the results of this experiment support the interference hypothesis.

---

> > ### Author Response · Authors · 2025-11-25
> > **Attention-span and pruning**
> >
> > **Attention Span**
> >
> > To test whether the degradation is due to limitations of attention-span, we kept the haystack fixed, but systematically changed the position of the query-recalled system within the haystack in Fig 6. When testing on a haystack with different systems, the different lightly-shaded dots in Fig 7.a show the performance of 2-after query and 3-after query degrades for all positions as more distinct systems are added to the haystack. This supports the hypothesis that the degradation is due to interference among haystack systems. This is not to say that there isn’t an impact of query-position as well — there is as one can see in Fig 6 where the MSE does depend slightly on which position is being recalled. However, there is more than just attention-span involved.
> >
> > To recap, our out-of-distribution experiments clearly show that whatever mechanism(s) the model is using to perform the 2-after-query and 3-after-query tasks, this mechanism is (a) not paying attention to the content of the query SPL; (b) subject to interference and degraded performance from the number of distinct systems in the haystack in a way that is different from how the 1-after-query recall – which necessarily must use the content of the query SPL — behaves.
> >
> > **Pruning analysis:**
> >
> > In our original submission, we had run one pruning experiment with only one random seed using essentially the default settings and had found 0% edge overlap between the circuit that did the 1-after-query task and that which did the 2-after-query task. We had originally done this to see whether there was any mechanistic plausibility to what we were seeing in our out-of-distribution experiments — the seeming presence of entirely distinct mechanisms for performing tasks that we had considered quite related to each other. In response to the reviews, we reran the pruning as we swept through different values for the optimization knob that controls sparsity: letting the resulting sparsity range across two orders of magnitude from 1000 to 10 edges in the final circuit.
> >
> > Then, we looked at the edge overlaps (visualized in Fig 8 as a heatmap) within those found for the 1-after-query task, within the 2-after-query task, and then across the 1-after-query task and the 2-after-query task. The level of overlap within 1-after-query circuits (found at different levels of sparsity using fresh runs with different randomness) was very high. Similarly, the level of overlap within 2-after-query circuits was also very high. However, the level of overlap across any 1-after-query circuit and any 2-after-query circuit was very low (although not always exactly zero). Within our paper’s story, the edge-pruning results are just saying that indeed it is possible to approximately do these tasks using non-overlapping circuits in the learned network. It is the out-of-distribution experiments that are actually convincingly showing that the mechanisms are different from each other because they behave differently. We’ll make this clearer in the final version.

---

### Meta-Review · Area_Chair_1pb1 · 2026-01-01

**Summary:**

The paper aims to explain the phenomenon of in-context learning (ICL) in transformers. Concretely, the paper introduces a synthetic "toy" problem designed to probe in-context learning (ICL) and associative recall. The setup involves interleaving segments of observations from deterministic linear dynamical systems. The study claims to identify two distinct mechanisms: a "label-based" mechanism used to recall the correct system at the start of a segment, and an "observation-based" mechanism used to predict subsequent tokens. The authors report that the observation-based mechanism emerges earlier in training and validate these findings using a translation task.

**Reviewer Concerns:**

The reviewers raised various concerns in their original reviews:

1. The validity of the toy setup is a premier concern expressed by the reviewers. In addition, the validity of the exact two circuit decomposition that is supported in the paper is something that reviewers are also not clear about.
2. Misunderstanding some of the experiments, e.g. sec. 4.
3. The relevance to transformers and LLMs themselves. The Reviewer sGTe expressed a concern on whether the proposed setting is related to practical LLMs and transformers.
4. Some writing issues, see Reviewer Rust comments.
5. Lack of experiments for supporting the theoretical claims.

**Reviewer Scores:**

I do believe the rebuttal attempts to offer an answer to the questions raised, but I am not sure whether the reviewers would agree. Firstly, the responses were pasted on 25th November, which would anyway leave limited time for interactions. In addition, I agree with the reviewers' concerns whether this proposed toy task indeed correlates with what the transformers and ICL does in practice. As the reviewer Rust mentions additional experimentation would be required for a thorough validation. Having said that, I do believe this is a promising direction, but it does not seem to pass the bar for ICLR yet.

---

### Decision · Program_Chairs · 2026-01-26

Reject